# DISENTANGLED PSEUDO-LABELING AND CLASSIFICATION FOR CLASS-IMBALANCED SEMI-SUPERVISED LEARNING

## ABSTRACT

Although significant improvements have been made in addressing class-imbalanced semi-supervised learning (CISSL), many algorithms still suffer from confirmation bias. Inaccurate pseudo-labels hinder the learning of the classifier, which in turn leads to further inaccurate pseudo-labels—creating a self-reinforcing loop that amplifies bias, particularly toward majority classes. This bias arises because the classifier that generates pseudo-labels is simultaneously trained on the unlabeled data it labels. To address this issue, we propose a novel CISSL algorithm, *Disentangled Pseudo-Labeling and Classification (DPC)*. DPC introduces an auxiliary classifier, dedicated solely to generating pseudo-labels, called a pseudo-labeler, which is attached to the representation layer of the backbone semi-supervised learning algorithm. To prevent confirmation bias, the pseudo-labeler is trained exclusively on labeled data, ensuring that pseudo-label generation remains unaffected by noisy unlabeled samples. Furthermore, to mitigate imbalanced feature representations—which are often biased toward majority classes and exacerbate confirmation bias—DPC propagates the classifier's training loss to the shared representation layer to encourage balanced feature learning. Benefiting from high-quality pseudo-labels and balanced feature representations, DPC achieves state-of-the-art classification performance on CISSL benchmark datasets.

## 1 INTRODUCTION

Semi-supervised learning (SSL) algorithms have achieved remarkable success by effectively leveraging large amounts of unlabeled data (Sohn et al., 2020; Berthelot et al., 2020; Chen et al., 2023; Wang et al., 2023b). A key factor driving their performance is the quality of pseudo-labels assigned to unlabeled samples, which serve as supervision during training. However, in class-imbalanced scenarios, pseudo-labels tend to be biased toward majority classes due to the skewed distribution of training samples. This bias often leads to a self-reinforcing loop in which the algorithm increasingly favors majority classes, further amplifying the imbalance in subsequent pseudo-labels. This phenomenon, known as confirmation bias, degrades the quality of pseudo-labels and overall performance. Therefore, alleviating class imbalance to enhance pseudo-label quality remains a critical challenge in class-imbalanced semi-supervised learning (CISSL).

Recently, various CISSL algorithms such as DARP Kim et al. (2020), ABC Lee et al. (2021), CoSSL Fan et al. (2022), SAW Lai et al. (2022), TCBC Li et al. (2024), CDMAD Lee and Kim (2024), and RECD Park et al. (2024) have been proposed to mitigate the bias toward majority classes. However, existing CISSL algorithms still suffer from confirmation bias, as illustrated in Fig. 1. This is primarily because they utilize unlabeled data in training the classifier, which is also responsible for generating pseudo-labels. The figure presents the confusion matrices of pseudo-labels for CIFAR-10-LT Cui et al. (2019a) under $\gamma_l = 100$ and $\gamma_u = 1$, where each entry $(i, j)$ indicates the proportion of samples from the $i$th class that are classified as the $j$th class. Even state-of-the-art algorithms such as CDMAD Lee and Kim (2024) exhibit difficulty in generating accurate pseudo-labels for minority class samples, highlighting the persistent confirmation bias toward majority classes.

To address confirmation bias, we propose a novel CISSL algorithm, termed *Disentangled Pseudo-Labeling and Classification (DPC)*. The core idea of DPC introduces an auxiliary classifier, dedicated

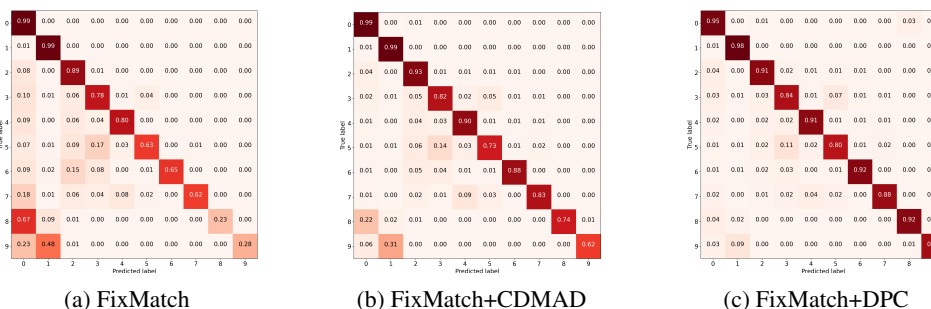

(a) FixMatch      (b) FixMatch+CDMAD      (c) FixMatch+DPC

Figure 1: Confusion matrices of class predictions for pseudo-labels on the unlabeled set of CIFAR-10-LT under $\gamma_l = 100$ and $\gamma_u = 1$.

solely to generating pseudo-labels, called a *pseudo-labeler* (PSL), which is trained exclusively on labeled data. It does not utilize any unlabeled samples during training, thereby avoiding the feedback loop that typically contributes to confirmation bias. PSL is responsible for generating pseudo-labels for the unlabeled samples used in both the backbone SSL algorithm and the *classifier* (CLS) of DPC. These high-quality pseudo-labels, obtained without being corrupted by unlabeled supervision, in turn facilitate more reliable training of both modules. Experimental results (Table 2 and Table 4) demonstrate that PSL produces more accurate pseudo-labels and contributes to achieving strong performance even under limited labeled data.

To mitigate class imbalance, both PSL and CLS adopt a simple reweighting scheme, where training samples are weighted inversely proportional to the number of samples in their respective classes during loss computation. While this strategy assumes knowledge of class distributions, the distribution of the unlabeled set is typically unknown in practice. By alleviating confirmation bias, PSL improves the quality of pseudo-labels, which in turn enables more accurate estimation of the unlabeled class distribution, as shown in Fig. 4. As a result, PSL enhances the effectiveness of the reweighting scheme even under distribution mismatch between the labeled and unlabeled sets. Despite employing this basic rebalancing strategy, DPC achieves superior performance, demonstrating the effectiveness of the proposed disentangled framework.

In addition to balancing label prediction, DPC also addresses feature representation imbalance, a common issue in SSL models trained on imbalanced datasets without explicit debiasing. Prior studies Chu et al. (2020); Kang et al. (2021); Park et al. (2024) have shown that feature maps often exhibit imbalance, with clusters corresponding to minority classes typically being sparser and more likely to overlap with those of other classes. To mitigate this issue, DPC propagates the classifier's reweighted loss—emphasizing minority classes—back to the shared representation layer of the backbone SSL model. This encourages the model to restructure the feature space, reducing the dominance of majority class features and improving the separability of minority class representations. The overall training scheme thus jointly promotes balance in both the classifier and representation space.

Experiments on benchmark datasets—CIFAR-10-LT Cui et al. (2019a), CIFAR-100-LT Cui et al. (2019a), and STL-10-LT Kim et al. (2020)—demonstrate that DPC consistently outperforms existing baseline methods across various settings. In-depth analyses confirm that DPC effectively mitigates confirmation bias at both the classifier and representation levels. Additionally, ablation studies reveal that each component of DPC contributes meaningfully to its overall performance. The source code for DPC is provided in the supplementary materials.

## 2 RELATED WORKS

To address class imbalance in SSL settings with both labeled and unlabeled data, various CISSL methods have been proposed. ABC Lee et al. (2021) and CoSSL Fan et al. (2022) introduce auxiliary classifiers to rebalance training losses based on the class distribution of labeled data. CReST Wei et al. (2021) progressively expands the labeled dataset by preferentially assigning pseudo-labels of minority classes to unlabeled samples. UDAL Lazarow et al. (2023) mitigates imbalance through a combination of logit adjustment and progressive distribution alignment. Although effective, many of these approaches rely on the assumption that the unknown class distribution of the unlabeled set matches that of the labeled set, which may limit their practicality.

Other methods aim to directly refine pseudo-labels or optimize training objectives. DARP Kim et al. (2020) refines biased pseudo-labels by formulating and solving a convex optimization problem. DASO Oh et al. (2022) generates pseudo-labels by integrating class predictions from both similarity-based and linear classifiers. SAW Lai et al. (2022) rebalances the training loss by adjusting weights based on measured class-wise learning difficulty. Adsh Guo and Li (2022) uses class-specific confidence thresholds to prioritize pseudo-labeling of minority class samples. InPL Yu et al. (2023) utilizes energy scores to identify well-predicted unlabeled instances that are likely to belong to minority classes.

Recent methods further advance debiasing strategies. L2AC Wang et al. (2023a) alleviates training bias using a bias-aware classifier that incorporates a bias attractor and a linear decision head. DePL Wang et al. (2022) addresses pseudo-label bias using counterfactual techniques and adaptive marginal loss. ACR Wei and Gan (2023) adaptively adjusts the training loss by estimating the class distribution of the unlabeled set through alignment between class predictions and distribution prototypes. TCBC Li et al. (2024) iteratively estimates class distributions from training data to mitigate class bias and refine pseudo-labels. CDMAD Lee and Kim (2024) measures classifier bias using solid color images and refines pseudo-labels based on the measured bias. RECD Park et al. (2024) estimates the class distribution of unlabeled samples and enhances representation quality by densifying minority class feature clusters. While these methods achieve promising results in CISSL, most remain susceptible to confirmation bias due to the classifier's dual role in both training and pseudo-label generation.

## 3 PRELIMINARIES

### 3.1 PROBLEM SETTINGS

We consider a $K$-class classification problem with a training dataset composed of a labeled set $\mathcal{X} = \{(x_n, y_n) : n \in (1, \ldots, N)\}$ and an unlabeled set $\mathcal{U} = \{(u_m) : m \in (1, \ldots, M)\}$, where each labeled sample $x_n \in \mathbb{R}^d$ is associated with its corresponding label $y_n \in \mathbb{R}^K$, and $u_m \in \mathbb{R}^d$ denotes the $m$th unlabeled sample. We denote by $N_k$ and $M_k$ the number of labeled and unlabeled samples of class $k$, respectively, with $\sum N_k = N$ and $\sum M_k = M$. For notational convenience, we assume that the labeled class sizes are sorted in descending order, i.e., $N_1 \geq N_2 \geq \ldots \geq N_K$. However, since the class distribution of the unlabeled set is unknown, we make no assumptions about the ordering of $M_k$. We define the class imbalance ratios of the labeled and unlabeled sets as $\gamma_l = \frac{N_1}{N_K}$ and $\gamma_u = \frac{\max M_k}{\min M_k}$, respectively. In this paper, we consider both scenarios where $\gamma_l = \gamma_u$ and where $\gamma_l \neq \gamma_u$. During training, we construct minibatches $\mathcal{B}_\mathcal{X} = \{(x_b^m, y_b^m) : b \in (1, \ldots, B_l)\} \subset \mathcal{X}$ and $\mathcal{B}_\mathcal{U} = \{(u_b^m) : b \in (1, \ldots, Bu)\} \subset \mathcal{U}$ for each iteration, where $B_l$ and $B_u$ denote the minibatch sizes for the labeled and unlabeled data, respectively. Using $\mathcal{B}_\mathcal{X}$ and $\mathcal{B}_\mathcal{U}$, our objective is to train a model $f_\theta(x) = \arg\max \phi(\xi(x)) : \mathbb{R}^d \to \mathbb{R}^K$ that accurately predicts the class labels of test samples $x_{test} \in \mathbb{R}^d$, where $\theta$ represents the model parameters, $\xi$ is a feature extractor, and $\phi$ mpas features to class probabilities.

### 3.2 BACKBONE SSL ALGORITHMS

We employed FixMatch Sohn et al. (2020) and ReMixMatch Berthelot et al. (2020) as backbone SSL algorithms for DPC, following previous CISSL algorithms (Kim et al., 2020; Lee et al., 2021; Lai et al., 2022; Fan et al., 2022; Lee and Kim, 2024; Park et al., 2024). Both methods employ consistency regularization Sajjadi et al. (2016); Miyato et al. (2018) by augmenting unlabeled samples through weak data augmentation $\alpha(\cdot)$ (e.g., flipping and cropping) and strong data augmentation $\mathcal{A}(\cdot)$ (e.g., RandomAugment Cubuk et al. (2020) and Cutout DeVries and Taylor (2017)). We denote the loss of the backbone SSL algorithm as $L_{back}$.

Specifically, FixMatch generates pseudo-labels for unlabeled samples $u_b^m$ as $\hat{q}_b^m = f_\theta(u_b^m)$, and calculates the consistency regularization loss by measuring the distance between the pseudo-labels and the predictions of strongly augmented samples $\mathcal{A}(u_b^m)$, provided that the model's confidence exceeds a threshold $\tau$, i.e., $\max(\phi_{back}(\xi(u_b^m))) \geq \tau$, as follows:

$$L_{con} = \frac{1}{B_u} \sum_{u_b^m \in \mathcal{B}_\mathcal{U}} \mathbf{I}(\max(\phi_{back}(\xi(q_b^m))) \geq \tau)\mathbf{H}(\phi_{back}(\xi(y|\mathcal{A}(u_b^m))), \hat{q}_b^m), \quad (1)$$

where $\mathbf{H}(\cdot, \cdot)$, $\mathbf{I}(\cdot)$, and $\phi_{back}$ denote the cross-entropy loss, indicator function, and backbone classifier, respectively. Additionally, FixMatch compares the class predictions for weakly augmented samples $\phi_{back}(\xi(\alpha(x_b^m)))$ against the ground-truth labels $y_b^m$, and calculates the classification loss as follows:

$$L_{back\_cls} = \frac{1}{B_l} \sum_{x_b^m \in \mathcal{B}_{\mathcal{X}}} \mathbf{H}(\phi_{back}(\xi(y|\alpha(x_b^m))), p_b^m), \tag{2}$$

where $p_b^m$ is the one-hot label of $y_b^m$. The total training loss used in FixMatch is then given by $L_{back} = L_{back\_cls} + L_{con}$.

ReMixMatch generates pseudo-labels for unlabeled samples $\alpha(u_b^m)$ as $\bar{q}_b^m$ by applying distribution alignment and sharpening to $q_b^m = \phi_{back}(\xi(\alpha(u_b^m)))$. It then computes MixUp regularization Zhang et al. (2018); Verma et al. (2022); Berthelot et al. (2019) and consistency regularization losses using $\bar{q}_b^m$, as follows:

$$L_{mix} = \frac{1}{B_l} \sum_{x_b^{mix} \in \mathcal{X}_{mix}} \mathbf{H}(\phi_{back}(\xi(y|x_b^{mix})), p_b^{mix}) + \frac{1}{B_u} \sum_{u_b^{mix} \in \mathcal{U}_{mix}} \mathbf{H}(\phi_{back}(\xi(y|u_b^{mix})), \bar{q}_b^{mix}),$$
$$\tag{3}$$

$$L_{con} = \frac{1}{B_u} \sum_{u_b^m \in \mathcal{B}_{\mathcal{U}}} \mathbf{H}(\phi_{back}(\xi(y|\mathcal{A}(u_b^m))), \bar{q}_b^m), \tag{4}$$

where $\mathcal{X}_{mix}$ and $\mathcal{U}_{mix}$ are MixUp-augmented batches of $\mathcal{B}_{\mathcal{X}}$ and $\mathcal{B}_{\mathcal{U}}$, respectively. Here, $x_b^{mix}$, $p_b^{mix}$, $u_b^{mix}$, and $\bar{q}_b^{mix}$ denote the mixed labeled sample, its label, the mixed unlabeled sample, and its pseudo-label, respectively. Additionally, ReMixMatch incorporates a self-supervised rotation loss Gidaris et al. (2018), defined as:

$$L_{rot} = \frac{1}{B_u} \sum_{u_b^m \in \mathcal{B}_{\mathcal{U}}} \mathbf{H}(f_{\theta_r}(r|\mathcal{R}(u_b^m, \hat{r})), \hat{r}), \tag{5}$$

where $\mathcal{R}(u_b^m, \hat{r})$ denotes the image $u_b^m$ rotated by $\hat{r}$ degrees, and $f_{\theta_r}$ returns the prediction for the rotation angle of $\mathcal{R}(u_b^m, \hat{r})$ using parameters $\theta_r$. The final training loss of ReMixMatch is $L_{back} = L_{mix} + L_{con} + L_{rot}$.

# 4 METHODOLOGY

## 4.1 PSEUDO-LABELER (PSL)

As described in Section 1, PSL generates pseudo-labels for unlabeled training samples, which are then used to train both the backbone SSL algorithm and the CLS as $q_{psl} = \phi_{psl}(\xi(y|\alpha(u_b^m)))$, where $\phi_{psl}$ denotes the PSL. Trained solely on the labeled set, PSL predicts class probabilities for unlabeled samples, which are unseen during its training, thereby mitigating confirmation bias.

To address class imbalance in the labeled set, PSL rebalances the training loss by assigning sample-wise weights inversely proportional to the number of samples in each sample's class, as follows:

$$L_{psl} = \frac{1}{B_l} \sum_{x_b^m \in \mathcal{M}\mathcal{B}_{\mathcal{X}}} \mathbf{W}(x_b^m) \mathbf{H}(\phi_{psl}(\xi(y|\alpha(x_b^m))), p_b^m), \tag{6}$$

where $\mathbf{W}(x_b^m) = \frac{N_K}{N_{y_b^m}}$. PSL is attached to the representation layer of the backbone SSL algorithm and is trained using the features extracted from this layer. To prevent interference with representation learning, gradient flow from PSL to the representation layer is detached.

DPC estimates the unknown class distribution of the unlabeled set by accumulating pseudo-labels predicted by PSL. At each training step $t$, this estimate is efficiently updated using an exponential moving average (EMA) as follows:

$$p_u^{(t)}(y) = \lambda p_u^{(t-1)}(y) + (1 - \lambda) \frac{1}{B_u} \sum_{u_b^m \in \mathcal{B}_{\mathcal{U}}} q_{psl}, \tag{7}$$

where $\lambda$ is the momentum coefficient of the EMA update and $p_u^{(0)}$ is initialized with the class distribution of the labeled set. CLS then utilizes the estimated class distribution $p_u^{(t)}(y)$ to rebalance the training loss on the unlabeled set at each training step $t$.

## 4.2 CLASSIFIER (CLS)

CLS is responsible for predicting class probabilities of input samples. To address class imbalance during training, it assigns a weight to each sample's training loss, inversely proportional to the frequency of its class in the training set. Its training loss on labeled samples, $L_{lcls}$, is similar to $L_{psl}$ as described in Eq. equation 6, but uses the distinct classifier $\phi_{cls}$ instead of $\phi_{psl}$, as follows:

$$L_{lcls} = \frac{1}{B_l} \sum_{x_b^m \in \mathcal{B}_\mathcal{X}} \mathbf{W}(x_b^m)\mathbf{H}(\phi_{cls}(\xi(y|\alpha(x_b^m))), p_b^m). \tag{8}$$

To rebalance the training loss for the unlabeled samples, $L_{ucls}$, CLS uses the pseudo-label $\hat{q}_{psl} = \arg\max q_{psl}$ and the class distribution of the unlabeled set estimated by PSL, as follows:

$$L_{ucls} = \frac{1}{B_u} \sum_{u_b^m \in \mathcal{B}_\mathcal{U}} \sum_{i=1}^{2} \mathbf{M}(u_b^m)\mathbf{W}(u_b^m)\mathbf{H}(\phi_{cls}(\xi(y|\alpha_i(u_b^m))), \tilde{q}_{psl}), \tag{9}$$

where $\mathbf{W}(u_b^m) = \frac{\min p_u(y)}{p_u(y)_{\hat{q}_{psl}}}$, and $\tilde{q}_{psl}$ denotes the one-hot encoding of the pseudo-label. Since majority classes tend to have lower pseudo-label precision under class imbalance, DPC mitigates this by masking low-confidence samples predicted as majority classes (top 50% most frequent) using mask $\mathbf{M}(\cdot)$ as follows:

$$\mathbf{M}(u_b^m) = \mathbf{I}(\max(q_{psl}) > \rho_{\hat{q}_{psl}}), \tag{10}$$

where $\rho_k$ is set to the average pseudo-label confidence for class $k$, while for minority classes—which typically have higher precision—we set $\rho_k = 0$ to retain their reliable predictions.

Furthermore, the backbone SSL algorithm learns feature representations directly from an imbalanced dataset without any debiasing, resulting in feature maps biased toward majority classes. To mitigate this inherent bias, DPC propagates the training loss to the representation layer as a feature loss, which is reweighted by the square of the sample-wise weight, $\mathbf{W}(\cdot)^2$. Squaring the weight induces a loss bias toward the minority classes, with the same degree as the class imbalance in the training set. When this squared-weighted loss is propagated to the representation layer, it offsets the original imbalance-induced bias in the feature maps, thereby achieving effective feature-level debiasing. The feature loss is calculated as follows:

$$L_{lfeature} = \frac{1}{B_l} \sum_{x_b^m \in \mathcal{B}_\mathcal{X}} (\mathbf{W}(x_b^m))^2 \mathbf{H}(\phi_{cls}(\xi(y|\alpha(x_b^m))), p_b^m), \tag{11}$$

$$L_{ufeature} = \frac{1}{B_u} \sum_{u_b^m \in \mathcal{B}_\mathcal{U}} \sum_{i=1}^{2} \mathbf{M}(u_b^m)(\mathbf{W}(u_b^m))^2 \mathbf{H}(\phi_{cls}(\xi(y|\alpha_i(u_b^m))), \tilde{q}_{psl}), \tag{12}$$

where $L_{lfeature}$ and $L_{ufeature}$ represent the feature losses of labeled samples and unlabeled samples, respectively. The training loss for mitigating feature map imbalance is denoted as $L_{feature} = L_{lfeature} + L_{ufeature}$. This loss does not update the CLS parameters but instead exclusively contributes to optimizing the shared feature extractor, thereby enhancing representation learning. The final training loss for CLS is as follows:

$$L_{cls} = L_{lcls} + L_{ucls} + \eta L_{feature}, \tag{13}$$

where $\eta$ is a hyperparameter. We fixed $\eta = 0.25$ across all experiments, which demonstrated the robustness of our algorithm. Moreover, in Appendix G, we conducted additional experiments by varying $\eta$ and observed consistently strong performance, indicating that our algorithm is not sensitive to the choice of this hyperparameter $\eta$.

## 4.3 END-TO-END TRAINING

Fig. 2 presents the training process of DPC using labeled and unlabeled data. The overall training loss of DPC is given by:

$$L_{DPC} = L_{back} + L_{psl} + L_{cls}. \tag{14}$$

We slightly modify the backbone SSL algorithm to utilize pseudo-labels generated by PSL for unlabeled samples. By leveraging more accurate pseudo-labels, the backbone learns higher-quality representations, thereby enhancing overall performance under class imbalance. The pseudo-code for the training process of DPC is provided in Appendix F.

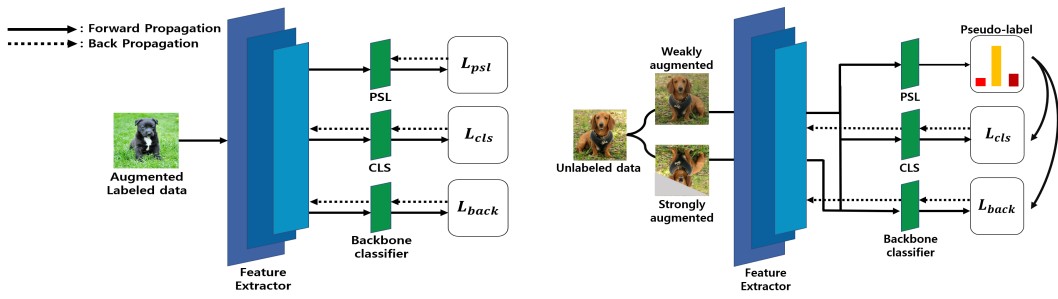

Figure 2: Training of DPC

# 5 EXPERIMENTS

## 5.1 EXPERIMENTAL SETUP

We performed experiments on CIFAR-10-LT Cui et al. (2019a), CIFAR-100-LT Krizhevsky (2009), and STL10-LT (Kim et al., 2020). Dataset details are provided in Appendix B. For evaluation, we measured the classification performance using balanced accuracy (bACC) Huang et al. (2016) and geometric mean (GM) (Kubat et al., 1997). The performance of the proposed algorithm, DPC, was compared with several CIL algorithms JAPKOWICZ (2000); Cao et al. (2019); Kang et al. (2020); Menon et al. (2021), SSL algorithms Sohn et al. (2020); Berthelot et al. (2020) and CISSL algorithms (Kim et al., 2020; Wei et al., 2021; Lee et al., 2021; Fan et al., 2022; Guo and Li, 2022; Wang et al., 2022; Lazarow et al., 2023; Wang et al., 2023a; Lai et al., 2022; Lee and Kim, 2024; Park et al., 2024). Additional training setup details are provided in Appendix B.

## 5.2 EXPERIMENTAL RESULTS

Table 1 summarizes the classification performances of the baseline algorithms and DPC on CIFAR-10-LT under the setting where $\gamma_l = \gamma_u$. DPC outperformed all baseline algorithms by large margins, even though DPC employed a basic imbalance mitigation strategy— reweighting samples inversely proportional to their class frequencies. These results are attributed to the effectiveness of DPC in alleviating confirmation bias, which enables the assignment of more accurate pseudo-labels, particularly for minority class samples. This improved pseudo-label quality significantly enhances performance, particularly for minority classes, as demonstrated in the detailed analysis.

Table 1: Comparison of bACC/GM on CIFAR-10-LT under $\gamma = \gamma_l = \gamma_u$

| CIFAR-10-LT ($\gamma = \gamma_l = \gamma_u$) | | | |
|---|---|---|---|
| Algorithm | $\gamma = 50$ | $\gamma = 100$ | $\gamma = 150$ |
| Vanilla (Cross-Entropy Loss) | $65.2_{\pm0.05}/61.1_{\pm0.09}$ | $58.8_{\pm0.13}/58.2_{\pm0.11}$ | $55.6_{\pm0.43}/44.0_{\pm0.98}$ |
| Re-sampling | $64.3_{\pm0.48}/60.6_{\pm0.67}$ | $55.8_{\pm0.47}/45.1_{\pm0.30}$ | $52.2_{\pm0.05}/38.2_{\pm1.49}$ |
| LDAM-DRW | $68.9_{\pm0.07}/67.0_{\pm0.08}$ | $62.8_{\pm0.17}/58.9_{\pm0.60}$ | $57.9_{\pm0.20}/50.4_{\pm0.30}$ |
| cRT | $67.8_{\pm0.13}/66.3_{\pm0.15}$ | $63.2_{\pm0.45}/59.9_{\pm0.40}$ | $59.3_{\pm0.10}/54.6_{\pm0.72}$ |
| FixMatch | $79.2_{\pm0.33}/77.8_{\pm0.36}$ | $71.5_{\pm0.72}/66.8_{\pm1.51}$ | $68.4_{\pm0.15}/59.9_{\pm0.43}$ |
| FixMatch+DARP+cRT | $85.8_{\pm0.43}/85.6_{\pm0.56}$ | $82.4_{\pm0.26}/81.8_{\pm0.17}$ | $79.6_{\pm0.42}/78.9_{\pm0.35}$ |
| FixMatch+CReST+LA | $85.6_{\pm0.36}/81.9_{\pm0.45}$ | $81.2_{\pm0.70}/74.5_{\pm0.99}$ | $71.9_{\pm2.24}/64.4_{\pm1.75}$ |
| FixMatch+ABC | $85.6_{\pm0.26}/85.2_{\pm0.29}$ | $81.1_{\pm1.14}/80.3_{\pm1.29}$ | $77.3_{\pm1.25}/75.6_{\pm1.65}$ |
| FixMatch+CoSSL | $86.8_{\pm0.30}/86.6_{\pm0.25}$ | $83.2_{\pm0.49}/82.7_{\pm0.60}$ | $80.3_{\pm0.55}/79.6_{\pm0.57}$ |
| FixMatch+SAW+LA | $86.2_{\pm0.15}/83.9_{\pm0.35}$ | $80.7_{\pm0.15}/77.5_{\pm0.21}$ | $73.7_{\pm0.06}/71.2_{\pm0.17}$ |
| FixMatch+Adsh | $83.4_{\pm0.06}/$ - | $76.5_{\pm0.35}/$ - | $71.5_{\pm0.30}/$ - |
| FixMatch+DebiasPL | -/ - | $80.6_{\pm0.50}/$ - | -/ - |
| FixMatch+UDAL | $86.5_{\pm0.29}/$ - | $81.4_{\pm0.39}/$ - | $77.9_{\pm0.33}/$ - |
| FixMatch+L2AC | -/ - | $82.1_{\pm0.57}/81.5_{\pm0.64}$ | $77.6_{\pm0.53}/75.8_{\pm0.71}$ |
| FixMatch+CDMAD | $87.3_{\pm0.12}/87.0_{\pm0.15}$ | $83.6_{\pm0.48}/83.1_{\pm0.57}$ | $80.8_{\pm0.86}/79.9_{\pm1.07}$ |
| FixMatch+RECD | $87.3_{\pm0.18}/87.2_{\pm0.19}$ | $84.0_{\pm0.13}/83.6_{\pm0.16}$ | $80.6_{\pm0.53}/79.7_{\pm0.66}$ |
| **FixMatch+DPC** | $\mathbf{87.7_{\pm0.15}/87.5_{\pm0.20}}$ | $\mathbf{84.5_{\pm0.36}/84.2_{\pm0.32}}$ | $\mathbf{81.8_{\pm0.60}/81.2_{\pm0.61}}$ |
| ReMixMatch | $81.5_{\pm0.26}/80.2_{\pm0.32}$ | $73.8_{\pm0.38}/69.5_{\pm0.84}$ | $69.9_{\pm0.47}/62.5_{\pm0.35}$ |
| ReMixMatch+DARP+cRT | $87.3_{\pm0.61}/87.0_{\pm0.11}$ | $83.5_{\pm0.07}/83.1_{\pm0.09}$ | $79.7_{\pm0.54}/78.9_{\pm0.49}$ |
| ReMixMatch+CReST+LA | $84.2_{\pm0.11}/$ - | $81.3_{\pm0.34}/$ - | $79.2_{\pm0.31}/$ - |
| ReMixMatch+ABC | $87.9_{\pm0.47}/87.6_{\pm0.51}$ | $84.5_{\pm0.32}/84.1_{\pm0.36}$ | $80.5_{\pm1.18}/79.5_{\pm1.36}$ |
| ReMixMatch+CoSSL | $87.7_{\pm0.21}/87.6_{\pm0.25}$ | $84.1_{\pm0.56}/83.7_{\pm0.66}$ | $81.3_{\pm0.83}/80.5_{\pm0.76}$ |
| ReMixMatch+SAW+cRT | $87.6_{\pm0.21}/87.4_{\pm0.26}$ | $85.4_{\pm0.32}/83.9_{\pm0.21}$ | $79.9_{\pm0.15}/79.9_{\pm0.12}$ |
| ReMixMatch+CDMAD | $88.1_{\pm0.13}/87.9_{\pm0.12}$ | $85.4_{\pm0.40}/85.2_{\pm0.41}$ | $82.5_{\pm0.33}/82.1_{\pm0.36}$ |
| ReMixMatch+RECD | $88.3_{\pm0.35}/88.1_{\pm0.35}$ | $85.5_{\pm0.46}/85.3_{\pm0.44}$ | $82.5_{\pm0.23}/82.0_{\pm0.30}$ |
| **ReMixMatch+DPC** | $\mathbf{88.7_{\pm0.23}/88.6_{\pm0.29}}$ | $\mathbf{85.9_{\pm0.21}/85.6_{\pm0.21}}$ | $\mathbf{83.0_{\pm0.32}/82.5_{\pm0.42}}$ |

Table 2 summarizes the classification performance of the baseline CISSL algorithms and DPC on CIFAR-100-LT under the setting where $\gamma_l = \gamma_u$. DPC consistently outperformed the baseline

algorithms by large margins. These results verify that DPC highly effective even on dataset with a large number of classes, such as CIFAR-100-LT, which contains 100 classes. Given that some classes have only one labeled sample under $\gamma = 100$, these results show the robustness of DPC in accurately assigning pseudo-labels despite extreme label scarcity.

Table 2: Comparison of bACC on CIFAR-100-LT under $\gamma = \gamma_l = \gamma_u$

| | CIFAR-100-LT ($\gamma = \gamma_l = \gamma_u$) | | |
|---|---|---|---|
| Algorithm | $\gamma = 20$ | $\gamma = 50$ | $\gamma = 100$ |
| FixMatch | $49.6_{\pm0.78}$ | $42.1_{\pm0.33}$ | $37.6_{\pm0.48}$ |
| FixMatch+DARP | $50.8_{\pm0.77}$ | $43.1_{\pm0.54}$ | $38.3_{\pm0.47}$ |
| FixMatch+DARP+cRT | $51.4_{\pm0.68}$ | $44.9_{\pm0.54}$ | $40.4_{\pm0.78}$ |
| FixMatch+CReST | $51.8_{\pm0.12}$ | $44.9_{\pm0.50}$ | $40.1_{\pm0.65}$ |
| FixMatch+CReST+LA | $52.9_{\pm0.07}$ | $47.3_{\pm0.17}$ | $42.7_{\pm0.70}$ |
| FixMatch+ABC | $53.3_{\pm0.79}$ | $46.7_{\pm0.26}$ | $41.2_{\pm0.06}$ |
| FixMatch+CoSSL | $53.9_{\pm0.78}$ | $47.6_{\pm0.57}$ | $43.0_{\pm0.61}$ |
| FixMatch+CDMAD | $54.3_{\pm0.44}$ | $48.8_{\pm0.75}$ | $44.1_{\pm0.29}$ |
| FixMatch+RECD | $54.6_{\pm0.36}$ | $47.8_{\pm0.17}$ | $42.8_{\pm0.40}$ |
| **FixMatch+DPC** | $\mathbf{55.5}_{\pm0.52}$ | $\mathbf{49.5}_{\pm0.21}$ | $\mathbf{44.5}_{\pm0.06}$ |
| ReMixMatch | $51.6_{\pm0.43}$ | $44.2_{\pm0.59}$ | $39.3_{\pm0.43}$ |
| ReMixMatch+DARP | $51.9_{\pm0.35}$ | $44.7_{\pm0.66}$ | $39.8_{\pm0.53}$ |
| ReMixMatch+DARP+cRT | $54.5_{\pm0.42}$ | $48.5_{\pm0.91}$ | $43.7_{\pm0.81}$ |
| ReMixMatch+CReST | $51.3_{\pm0.34}$ | $45.5_{\pm0.76}$ | $41.0_{\pm0.78}$ |
| ReMixMatch+CReST+LA | $51.9_{\pm0.60}$ | $46.6_{\pm1.14}$ | $41.7_{\pm0.69}$ |
| ReMixMatch+ABC | $55.6_{\pm0.35}$ | $47.9_{\pm0.10}$ | $42.2_{\pm0.12}$ |
| ReMixMatch+CoSSL | $55.8_{\pm0.62}$ | $48.9_{\pm0.61}$ | $44.1_{\pm0.59}$ |
| ReMixMatch+CDMAD | $55.8_{\pm0.46}$ | $51.1_{\pm0.42}$ | $44.9_{\pm0.42}$ |
| ReMixMatch+RECD | $55.9_{\pm0.36}$ | $49.5_{\pm0.21}$ | $43.5_{\pm0.27}$ |
| **ReMixMatch+DPC** | $\mathbf{57.7}_{\pm0.40}$ | $\mathbf{52.4}_{\pm0.17}$ | $\mathbf{46.1}_{\pm0.17}$ |

Table 3 summarizes the classification performance of the baseline algorithms and DPC on CIFAR-10-LT under the setting where $\gamma_l \neq \gamma_u$, and on STL-10-LT under unknown $\gamma_u$. DPC outperformed the baseline algorithms by large margins. These results demonstrate that DPC effectively mitigates confirmation bias even under $\gamma_l \neq \gamma_u$, especially when there are a large number of unlabeled samples in minority classes. Furthermore, DPC outperformed the baseline algorithms that use ReMixMatch* Kim et al. (2020), which incorporates class distribution estimation for distribution alignment, as the backbone SSL algorithm. This may be because DPC effectively estimates the unknown class distribution of the unlabeled set, owing to highly accurate pseudo-labels.

Table 3: Comparison of bACC/GM on CIFAR-10-LT and STL-10-LT under $\gamma_l \neq \gamma_u$.

| | CIFAR-10-LT ($\gamma_l = 100$) | | | STL-10-LT ($\gamma_u = $ Unknown) | |
|---|---|---|---|---|---|
| Algorithm | $\gamma_u = 1$ | $\gamma_u = 50$ | $\gamma_u = 150$ | $\gamma_l = 10$ | $\gamma_l = 20$ |
| FixMatch | $68.9_{\pm1.95}/42.8_{\pm8.11}$ | $73.9_{\pm0.25}/70.5_{\pm0.52}$ | $69.6_{\pm0.60}/62.6_{\pm1.11}$ | $72.9_{\pm0.09}/69.6_{\pm0.01}$ | $63.4_{\pm0.21}/52.6_{\pm0.09}$ |
| /+DARP | $85.4_{\pm0.55}/85.0_{\pm0.65}$ | $77.3_{\pm0.17}/75.5_{\pm0.21}$ | $72.9_{\pm0.24}/69.5_{\pm0.18}$ | $77.8_{\pm0.33}/76.5_{\pm0.40}$ | $69.9_{\pm1.77}/65.4_{\pm3.07}$ |
| /+DARP+LA | $86.6_{\pm1.11}/86.2_{\pm1.15}$ | $82.3_{\pm0.32}/81.5_{\pm0.29}$ | $78.9_{\pm0.23}/77.7_{\pm0.06}$ | $78.6_{\pm0.30}/77.4_{\pm0.40}$ | $71.9_{\pm0.49}/68.7_{\pm0.51}$ |
| /+DARP+cRT | $87.0_{\pm0.70}/86.8_{\pm0.67}$ | $82.7_{\pm0.21}/82.3_{\pm0.25}$ | $80.7_{\pm0.44}/80.2_{\pm0.61}$ | $79.3_{\pm0.23}/78.7_{\pm0.21}$ | $74.1_{\pm0.61}/73.1_{\pm1.21}$ |
| /+ABC | $82.7_{\pm0.49}/81.9_{\pm0.68}$ | $82.7_{\pm0.64}/82.0_{\pm0.76}$ | $78.4_{\pm0.87}/77.2_{\pm1.07}$ | $79.1_{\pm0.46}/78.1_{\pm0.57}$ | $73.8_{\pm0.15}/72.1_{\pm0.15}$ |
| /+SAW | $81.2_{\pm0.68}/80.2_{\pm0.91}$ | $79.8_{\pm0.25}/79.1_{\pm0.32}$ | $74.5_{\pm0.97}/72.5_{\pm1.37}$ | $78.3_{\pm0.25}/77.0_{\pm0.19}$ | $71.9_{\pm0.81}/69.0_{\pm0.81}$ |
| /+SAW+LA | $84.5_{\pm0.68}/84.1_{\pm0.78}$ | $82.9_{\pm0.38}/82.6_{\pm0.38}$ | $79.1_{\pm0.81}/78.6_{\pm0.91}$ | -/ - | -/ - |
| /+SAW+cRT | $84.6_{\pm0.23}/84.4_{\pm0.26}$ | $81.6_{\pm0.38}/81.3_{\pm0.32}$ | $77.6_{\pm0.40}/77.1_{\pm0.41}$ | -/ - | -/ - |
| /+CDMAD | $87.5_{\pm0.46}/87.1_{\pm0.50}$ | $85.7_{\pm0.36}/85.3_{\pm0.38}$ | $82.3_{\pm0.23}/81.8_{\pm0.29}$ | $79.9_{\pm0.23}/78.9_{\pm0.38}$ | $75.2_{\pm0.40}/73.5_{\pm0.31}$ |
| /+RECD | $90.2_{\pm0.57}/90.0_{\pm064}$ | $85.6_{\pm0.15}/85.3_{\pm0.15}$ | $82.3_{\pm0.39}/81.8_{\pm0.45}$ | $81.4_{\pm0.28}/80.6_{\pm0.38}$ | $79.0_{\pm0.48}/78.1_{\pm0.54}$ |
| **/+DPC** | $\mathbf{91.1}_{\pm0.15}/\mathbf{90.9}_{\pm0.15}$ | $\mathbf{86.2}_{\pm0.27}/\mathbf{86.0}_{\pm0.29}$ | $\mathbf{83.5}_{\pm0.23}/\mathbf{83.1}_{\pm0.23}$ | $\mathbf{81.6}_{\pm0.20}/\mathbf{81.0}_{\pm0.25}$ | $\mathbf{79.6}_{\pm0.60}/\mathbf{78.8}_{\pm0.61}$ |
| ReMixMatch* | $85.0_{\pm1.35}/84.3_{\pm1.55}$ | $77.0_{\pm0.12}/74.7_{\pm0.04}$ | $72.8_{\pm0.10}/68.8_{\pm0.21}$ | $76.7_{\pm0.15}/73.9_{\pm0.32}$ | $67.7_{\pm0.46}/60.3_{\pm0.76}$ |
| /+DARP | $86.9_{\pm0.10}/86.4_{\pm0.15}$ | $77.4_{\pm0.22}/75.0_{\pm0.25}$ | $73.2_{\pm0.11}/69.2_{\pm0.31}$ | $79.4_{\pm0.07}/78.2_{\pm0.10}$ | $70.9_{\pm0.44}/67.0_{\pm1.62}$ |
| /+DARP+LA | $81.8_{\pm0.45}/80.9_{\pm0.40}$ | $83.9_{\pm0.42}/83.4_{\pm0.45}$ | $81.1_{\pm0.20}/80.3_{\pm0.26}$ | $80.6_{\pm0.45}/79.6_{\pm0.55}$ | $76.8_{\pm0.60}/74.8_{\pm0.68}$ |
| /+DARP+cRT | $88.7_{\pm0.25}/88.5_{\pm0.26}$ | $83.5_{\pm0.53}/83.1_{\pm0.51}$ | $80.9_{\pm0.25}/80.3_{\pm0.31}$ | $80.9_{\pm0.53}/80.0_{\pm0.46}$ | $76.7_{\pm0.50}/74.9_{\pm0.70}$ |
| /+SAW | $87.0_{\pm0.75}/86.4_{\pm0.85}$ | $80.6_{\pm1.57}/79.2_{\pm2.19}$ | $77.6_{\pm0.76}/76.0_{\pm0.93}$ | $82.0_{\pm0.55}/81.0_{\pm0.64}$ | $79.2_{\pm0.44}/77.9_{\pm0.52}$ |
| /+SAW+LA | $74.2_{\pm1.49}/65.1_{\pm2.36}$ | $84.8_{\pm1.07}/82.4_{\pm2.32}$ | $81.3_{\pm2.42}/80.9_{\pm2.47}$ | -/ - | -/ - |
| /+SAW+cRT | $88.8_{\pm0.79}/88.6_{\pm0.83}$ | $84.5_{\pm0.78}/83.6_{\pm1.27}$ | $82.4_{\pm0.10}/82.0_{\pm0.10}$ | -/ - | -/ - |
| ReMixMatch | $48.3_{\pm0.14}/19.5_{\pm0.85}$ | $75.1_{\pm0.43}/71.9_{\pm0.77}$ | $72.5_{\pm0.10}/68.3_{\pm0.32}$ | $67.8_{\pm0.45}/61.1_{\pm0.92}$ | $60.1_{\pm1.18}/44.9_{\pm1.52}$ |
| /+ABC | $76.4_{\pm5.34}/74.8_{\pm6.05}$ | $85.2_{\pm0.20}/84.7_{\pm0.25}$ | $80.4_{\pm0.40}/80.0_{\pm0.44}$ | $76.8_{\pm0.52}/74.8_{\pm0.64}$ | $71.2_{\pm1.37}/67.4_{\pm1.89}$ |
| /+CDMAD | $89.9_{\pm0.45}/89.6_{\pm0.46}$ | $86.9_{\pm0.21}/86.7_{\pm0.17}$ | $83.1_{\pm0.46}/82.7_{\pm0.50}$ | $83.0_{\pm0.38}/82.1_{\pm0.35}$ | $81.9_{\pm0.32}/80.9_{\pm0.44}$ |
| /+RECD | $90.3_{\pm0.40}/90.2_{\pm0.41}$ | $86.8_{\pm0.17}/86.6_{\pm0.18}$ | $83.9_{\pm0.11}/83.7_{\pm0.12}$ | $84.9_{\pm0.41}/84.4_{\pm0.47}$ | $82.5_{\pm0.27}/81.7_{\pm0.31}$ |
| **/+DPC** | $\mathbf{91.0}_{\pm0.15}/\mathbf{90.8}_{\pm0.17}$ | $\mathbf{87.4}_{\pm0.12}/\mathbf{87.1}_{\pm0.17}$ | $\mathbf{84.3}_{\pm0.20}/\mathbf{84.0}_{\pm0.20}$ | $\mathbf{85.0}_{\pm0.32}/\mathbf{84.5}_{\pm0.38}$ | $\mathbf{83.1}_{\pm0.56}/\mathbf{82.3}_{\pm0.70}$ |

Algorithms such as DASO and ACR, whose classification performances were reported under different experimental settings in their original papers, are compared with DPC in Appendix C. In addition, we compare the fine-grained performance—across the "Many", "Medium", and "Few" groups, which correspond to data partitions based on class size—of FixMatch, ReMixatch, SAW, and SAW+cRT with that of DPC in Appendix D. Furthermore, to verify the compatibility of DPC with recent SSL

algorithms, we performed experiments using FreeMatch Wang et al. (2023b) as the backbone SSL algorithm in Appendix E.

## 5.3 DETAILED ANALYSES

We argued that DPC effectively mitigates confirmation bias and generates more accurate pseudo-labels by introducing PSL. To validate this, we compare pseudo-label and test accuracies with and without the use of PSL on CIFAR-10-LT and CIFAR-100-LT under the setting $\gamma = \gamma_l = \gamma_u$. Without PSL, pseudo-labels are generated by the CLS, which is jointly trained on the unlabeled set. As shown in Table 4, DPC produces much more precise pseudo-labels, resulting in improved test performance.

Table 4: Comparison of pseudo-label accuracy/test accuracy with and without the use of PSL on CIFAR-10-LT and CIFAR-100-LT under $\gamma = \gamma_l = \gamma_u$.

| | CIFAR-10-LT ($\gamma = \gamma_l = \gamma_u$) | | | CIFAR-100-LT ($\gamma = \gamma_l = \gamma_u$) | | |
|---|---|---|---|---|---|---|
| Imbalance ratio | $\gamma = 150$ | $\gamma = 100$ | $\gamma = 50$ | $\gamma = 100$ | $\gamma = 50$ | $\gamma = 20$ |
| Without PSL | 88.2 / 80.5 | 88.9 / 82.7 | 89.3 / 86.2 | 70.5 / 43.8 | 69.7 / 48.8 | 68.2 / 53.8 |
| **FixMatch+DPC** | **94.4 / 81.8** | **94.3 / 84.5** | **93.9 / 87.7** | **75.9 / 44.5** | **74.7 / 49.5** | **72.7 / 55.5** |

In addition, by effectively mitigating bias in the feature map using the feature loss in Eq. (11) and Eq. (12), DPC enables more distinct and disentangled feature representations for minority classes (red and yellow) within the unlabeled training set. Consequently, clusters corresponding to minority classes form more independently and exhibit minimal overlap with majority-class clusters (black and blue) in the feature map. Fig. 3 presents the T-SNE visualizations of the unlabeled training set of CIFAR-10-LT under the setting $\gamma_l = 100, \gamma_u = 1$. Fig. 3a presents the T-SNE results for FixMatch, where the feature maps show significant overlap between classes. Fig. 3b presents the T-SNE results for FixMatch+DPC without the use of the feature loss, where the overlap is reduced owing to PSL and CLS, but clusters remain somewhat entangled and decision boundaries are less clear. Fig. 3c presents the T-SNE results for FixMatch+DPC with the feature loss, demonstrating the combined effect of reducing confirmation bias in both pseudo-labeling and representation learning. Compared to Fig. 3a and Fig. 3b, the decision boundaries are more distinct, and the clusters—especially those of minority classes (red and yellow)—are more compact and well-separated. These results highlight DPC's effectiveness in enhancing class separation and learning better representations, particularly for underrepresented classes.

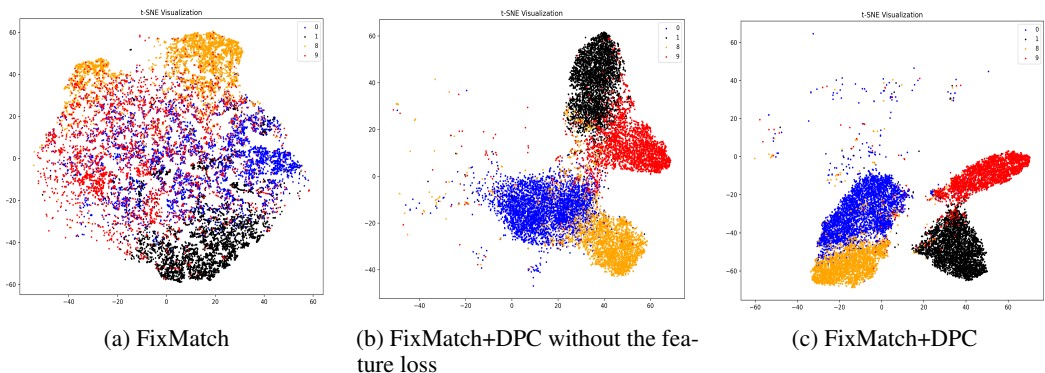

(a) FixMatch      (b) FixMatch+DPC without the feature loss      (c) FixMatch+DPC

Figure 3: T-SNE of the unlabeled training set on CIFAR-10-LT under $\gamma_l = 100, \gamma_u = 1$.

This separation is not only evident in the feature map but also reflected in the performance metrics, where the improvement in bACC highlights its contribution to robust learning under class-imbalanced scenarios. Specifically, FixMatch achieves a bACC of **68.9**, the variant without the feature loss achieves **85.6**, and FixMatch+DPC achieves **91.1**, demonstrating the significant impact of enhanced cluster separation. We compared the T-SNE visualizations of the test set on CIFAR-10-LT under the setting $\gamma_l = 100, \gamma_u = 1$ in Appendix H.

Furthermore, Fig. 4 presents the true class distribution of the training set and the estimated class distribution of the unlabeled set. When there is a significant difference between the distributions of

the labeled and unlabeled sets, pseudo-label predictions tend to be biased toward classes with more labeled data, thereby intensifying confirmation bias. As shown in Fig. 4, the proposed debiasing strategy enables accurate pseudo-label prediction even under severe distribution mismatch. This enhanced estimation allows the algorithm to handle unknown unlabeled distributions more robustly, contributing to better generalization in CISSL.

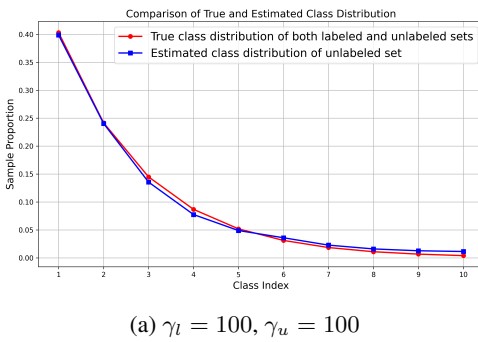
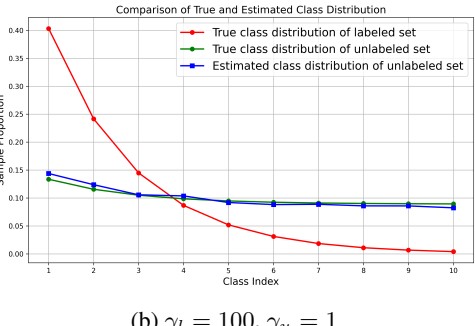

(a) $\gamma_l = 100, \gamma_u = 100$        (b) $\gamma_l = 100, \gamma_u = 1$

Figure 4: True class distributions of the labeled and unlabeled sets, along with the class distribution of the unlabeled set estimated by DPC. FixMatch+DPC was trained on CIFAR-10-LT with $\gamma_l = 100, \gamma_u = 100$ and $\gamma_l = 100, \gamma_u = 1$.

### 5.4 ABLATION STUDY

To investigate the effectiveness of each component of DPC, we conducted an ablation study on STL-10-LT under class distribution mismatch between the labeled and unlabeled sets (with unknown $\gamma_u$). Each row in Table 5 represents a variant of DPC with a specific component removed. We consider the following ablations: **(1)** *Without masking in Eq.* (10) *for CLS*, **(2)** *Without the feature loss in Eq.* (11) *and Eq.* (12) *for representation learning*, and **(3)** *Without PSL (i.e., without separating pseudo-label generation from classification)*. First, without masking for CLS allows samples misclassified as majority classes to influence training, which slightly degrades overall performance due to reinforced bias toward dominant classes. Second, excluding the feature loss in representation learning weakens debiasing at the representation level, which leads to performance degradation. Lastly, bypassing PSL and generating pseudo-labels directly from CLS degrades performance, as CLS—utilizing imbalanced unlabeled data for training —fails to mitigate confirmation bias effectively. These results verify the effectiveness of each component of DPC.

Table 5: Ablation study for DPC on STL-10-LT under $\gamma_l \neq \gamma_u$(unknown $\gamma_u$), where the evaluation metric is baCC/GM.

| Ablation study | $\gamma_l = 10$ | $\gamma_l = 20$ |
|---|---|---|
| Without masking for CLS | 80.7 / 80.0 | 79.3 / 78.4 |
| Without feature loss for representation learning | 80.5 / 79.8 | 78.8 / 77.9 |
| Without PSL in generating pseudo-label | 80.9 / 80.8 | 78.6 / 77.6 |
| **FixMatch+DPC (Proposed algorithm)** | **81.6 / 81.0** | **79.6 / 78.8** |

## 6 CONCLUSION

In this paper, we addressed confirmation bias in CISSL. We proposed DPC, a novel framework that separates pseudo-label generation from test sample prediction by employing two distinct modules: PSL and CLS. By training the PSL solely on labeled data, DPC avoids the self-reinforcing loop of incorrect pseudo-labels, thereby mitigating confirmation bias. Furthermore, DPC introduces a rebalanced loss on the shared feature representation, effectively correcting feature-level imbalance caused by the backbone SSL algorithm. Extensive experiments on various CISSL benchmarks demonstrate that DPC consistently outperforms existing methods, particularly under severe imbalance. These results highlight the importance of decoupling pseudo-labeling from classifier training and correcting representation-level bias to achieve robust and balanced learning in CISSL.

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

## A    ADDITIONAL RELATED WORKS

### A.1    CLASS-IMBALANCED SUPERVISED LEARNING

Class-imbalanced learning (CIL) techniques aim to train unbiased classifiers in scenarios where the training set exhibits severe class imbalance. A fundamental strategy is resampling Wang et al. (2015); Estabrooks et al. (2004); Liu et al. (2008); Chawla et al. (2002), which modifies the class distribution by oversampling minority class samples or undersampling majority ones. Another common approach is reweighting Ren et al. (2018); Li et al. (2014); Cui et al. (2019b); Lin et al. (2017), which adjusts the training loss by assigning higher weights to samples from minority classes. Beyond these, various methods have been proposed to further reduce class imbalance bias. LDAM Cao et al. (2019) introduces a loss function that incorporates label-distribution-aware margins, aiming to improve generalization for minority classes by explicitly expanding decision margins based on class frequencies. LA Menon et al. (2021) minimizes the balanced training loss by calibrating the logits for samples based on the class distribution of the training set. Decoupled learning Kang et al. (2020) separates representation learning from classifier training, resulting in a high-quality feature space and an unbiased classifier. More recently, approaches such as multi-expert learning Wang et al. (2021b); Zhou et al. (2020); Zhang et al. (2022); Xiang et al. (2020); Cai et al. (2021); Li et al. (2022a), contrastive learning Cui et al. (2021); Kang et al. (2021); Li et al. (2022b); Wang et al. (2021a), and knowledge distillation Iscen et al. (2021); Li et al. (2021) have been explored for CIL. While these methods have shown remarkable success in fully supervised settings, their reliance on labeled data limits their direct applicability in semi-supervised scenarios, where a substantial portion of the dataset remains unlabeled.

### A.2    SEMI-SUPERVISED LEARNING

Semi-supervised learning (SSL) has received increasing attention due to its ability to effectively leverage both labeled and unlabeled data. Two key techniques in SSL are entropy minimization and consistency regularization. Entropy minimization Grandvalet and Bengio (2004) encourages the model to avoid placing decision boundaries in high-density regions by reducing the predictive uncertainty on unlabeled samples. This is typically implemented through pseudo-labeling Lee (2013), where model predictions are converted into one-hot targets or sharpened probability distributions. Consistency regularization Sajjadi et al. (2016); Miyato et al. (2018); Tarvainen and Valpola (2017) enforces stability in classifier predictions under input perturbations, ensuring that decision boundaries are placed in low-density regions. Leading SSL algorithms such as MixMatch Berthelot et al. (2019), ReMixMatch Berthelot et al. (2020), and FixMatch Sohn et al. (2020) integrate consistency regularization with pseudo-labeling strategies. Additionally, MixMatch and ReMixMatch conduct Mixup regularization Zhang et al. (2018); Verma et al. (2022) to further enhance generalization. Recently, FlexMatch Zhang et al. (2021) accounts for the learning progress and difficulty levels across classes. MarginMatch Sosea and Caragea (2023) masks out pseudo-labels with low confidence, while SoftMatch Chen et al. (2023) assigns weights to pseudo-labels using a truncated Gaussian distribution. FreeMatch Wang et al. (2023b) extends FixMatch by introducing class-wise adaptive confidence thresholds and a fairness-oriented loss. HyperMatch Zhou et al. (2023) utilizes relaxed contrastive loss to mitigates confirmation bias. These recent SSL algorithms build on FixMatch Sohn et al. (2020), leveraging its simplicity and effectiveness.

## B    DETAILED DESCRIPTIONS FOR THE EXPERIMENTAL SETUP

### B.1    DATASETS

**CIFAR-10-LT** and **CIFAR-100-LT** are artificially constructed long-tailed variants of CIFAR-10 and CIFAR-100 Krizhevsky (2009), respectively, created by reducing the number of training samples in tail classes. The numbers of labeled and unlabeled samples for the $k$th class, denoted as $N_k$ and $M_k$, are given by $N_k = N_1 \times \gamma_l^{-\frac{k-1}{K-1}}$ and $M_k = M_1 \times \gamma_u^{-\frac{k-1}{K-1}}$, where $K$ is the total number of classes. For CIFAR-10-LT, we set $N_1 = 1500$ and $M_1 = 3000$. We first evaluate the performance under the setting $\gamma = \gamma_l = \gamma_u$, varying $\gamma$ by 50, 100, and 150. Additionally, we experiment under mismatched imbalance settings with $\gamma_l = 100$ and $\gamma_u$ set to 1, 50, and 150. For CIFAR-100-LT, we set $N_1 = 150$

and $M_1 = 300$, and conducted experiments with $\gamma = \gamma_l = \gamma_u$, varying $\gamma$ over 20, 50 and 100.

**STL-10-LT** is an artificially constructed long-tailed variant of STL-10 Coates et al. (2011) by reducing the number of training samples in tail classes. The number of labeled samples for the $k$th class, denoted as $N_k$, is given by $N_k = N_1 \times \gamma_l^{-\frac{k-1}{K-1}}$, where $K$ is the total number of classes. STL-10-LT contains 100,000 unlabeled samples with an unknown class distribution. We set $N_1 = 450$ and utilized the entire unlabeled set during training. We conducted experiments on STL-10-LT, varying $\gamma_l$ over 10 and 20.

## B.2 Training setups and baseline algorithms

We employed Wide ResNet-28-2 Zagoruyko and Komodakis (2016) as the deep convolutional neural network for all experiments. For optimization, we used Adam Kingma and Ba (2015), setting the learning rate to $1.5 \times 10^{-3}$ for FixMatch and $2 \times 10^{-3}$ for ReMixMatch. Exponential moving average (EMA) with a decay rate of 0.999 was applied to update the network parameters at each iteration. The weight for the feature loss, $\eta$, was set to 0.25. Each experiment was conducted three times with different random seeds, and we report the mean and standard deviation of the balanced accuracy (bACC) and geometric mean (GM) across runs. For CIFAR-100-LT, we applied a weight decay of 0.08 for FixMatch and 0.1 for ReMixMatch to mitigate overfitting due to the limited number of samples per class. For the remaining datasets, weight decay was set to 0.04 for FixMatch and 0.06 for ReMixMatch when the total number of training samples was below $3 \times 10^4$, and to 0.01 for FixMatch and 0.02 for ReMixMatch otherwise, considering the reduced overfitting risk with larger datasets. FixMatch was trained for 500 epochs, with each epoch consisting of 500 iterations. We set the labeled batch size to 32 and the unlabeled batch size to 64. For STL-10-LT, the unlabeled batch size was increased to 96. To leverage the entire unlabeled dataset, the confidence threshold $\tau$ was set to 0. ReMixMatch was trained for 300 epochs, with each epoch consisting of 500 iterations. The labeled batch size was set to 64, and the unlabeled batch size was set to 128. We did not employ the distribution alignment strategy introduced in ReMixMatch, as the true class distribution of the unlabeled data is unknown. Instead, we incorporated the supervised classification loss computed on weakly augmented labeled examples in the overall training loss. All experiments were conducted using NVIDIA RTX 3090 GPUs with PyTorch version 1.8.1. The source code is included in the supplementary material.

## C Comparison of DPC with DASO and ACR

Since the experimental settings used to evaluate DASO Oh et al. (2022) and ACR Wei and Gan (2023) differ slightly from ours, we refrained from directly comparing their reported performance with our results. To ensure a fair comparison, we evaluated DASO and ACR under the same experimental configurations as DPC. As shown in Table 6, Table 7, Table 8, and Table 9, DPC consistently outperformed both DASO and ACR. DPC achieved effective performance across diverse scenarios by effectively mitigating confirmation bias.

Table 6: Comparison of bACC/GM on CIFAR-10-LT under $\gamma = \gamma_l = \gamma_u$

| CIFAR-10-LT ($\gamma = \gamma_l = \gamma_u$) | | | |
|---|---|---|---|
| Algorithm | $\gamma = 50$ | $\gamma = 100$ | $\gamma = 150$ |
| FixMatch+DASO | 81.8/ 81.0 | 75.7/ 74.0 | 72.0/ 68.9 |
| FixMatch+DASO+LA | 84.1/ 83.7 | 79.4/ 78.8 | 76.5/ 75.5 |
| **FixMatch+DPC** | **87.7/ 87.5** | **84.5/ 84.2** | **81.8/ 81.2** |
| ReMixMatch+DASO | 82.5/ 81.9 | 76.0/ 73.9 | 70.8/ 66.5 |
| ReMixMatch+DASO+LA | 85.9/ 85.7 | 82.8/ 82.4 | 79.0/ 78.4 |
| **ReMixMatch+DPC** | **88.7/ 88.6** | **85.9/ 85.6** | **83.0/ 82.5** |

Table 7: Comparison of bACC/GM on CIFAR-100-LT under $\gamma = \gamma_l = \gamma_u$

| CIFAR-100-LT ($\gamma = \gamma_l = \gamma_u$) | | | |
|---|---|---|---|
| Algorithm | $r = 20$ | $r = 50$ | $r = 100$ |
| FixMatch+DASO | 45.8 | 39.2 | 33.9 |
| FixMatch+DASO+LA | 46.2 | 39.9 | 34.5 |
| **FixMatch+DPC** | **55.5** | **49.5** | **44.5** |
| ReMixMatch+DASO | 51.5 | 43.0 | 38.2 |
| ReMixMatch+DASO+LA | 52.8 | 45.5 | 40.3 |
| **ReMixMatch+DPC** | **57.7** | **52.4** | **46.1** |

Table 8: Comparison of bACC/GM on CIFAR-10-LT and STL-10-LT under $\gamma_l \neq \gamma_u$.

| | CIFAR-10-LT ($\gamma_l = 100$) | | | STL-10-LT ($\gamma_u$ = Unknown) | |
|---|---|---|---|---|---|
| Algorithm | $\gamma_u = 1$ | $\gamma_u = 50$ | $\gamma_u = 150$ | $\gamma_l = 10$ | $\gamma_l = 20$ |
| FixMatch+DASO | 86.4/ 86.0 | 79.1/ 78.2 | 74.2/ 71.6 | 68.4/ 65.3 | 62.1/ 58.9 |
| FixMatch+DASO+LA | 86.2/ 85.8 | 81.7/ 81.2 | 78.0/ 77.0 | 68.9/ 66.3 | 66.0/ 64.6 |
| **FixMatch+DPC** | **91.1/ 90.9** | **86.2/ 86.0** | **83.5/ 83.1** | **81.6/ 81.0** | **79.6/ 78.8** |
| ReMixMatch+DASO | 89.6/ 89.3 | 79.6/ 77.8 | 72.3/ 69.0 | 75.1/ 73.6 | 66.8/ 61.8 |
| ReMixMatch+DASO+LA | 80.6/ 77.7 | 84.8/ 84.5 | 79.7/ 79.2 | 78.1/ 77.3 | 75.3/ 74.0 |
| **ReMixMatch+DPC** | **91.0/ 90.8** | **87.4/ 87.1** | **84.3/ 84.0** | **85.0/ 84.5** | **83.1/ 82.3** |

Table 9: Comparison of bACC/GM on CIFAR-10-LT under $\gamma = \gamma_l = \gamma_u = 100$ and $\gamma_l = 100, \gamma_u = 1$

| Algorithm/ CIFAR-10-LT | $\gamma_l = \gamma_u = 100$ | $\gamma_l = 100, \gamma_u = 1$ |
|---|---|---|
| FixMatch+ACR | 81.8/ 81.4 | 85.6/ 85.3 |
| **FixMatch+DPC** | **84.5/ 84.2** | **91.1/ 90.9** |

## D   FINE-GRAINED EXPERIMENTAL RESULTS OF DPC

To evaluate the effectiveness of DPC in addressing class imbalance, we categorized the classes in CIFAR-10-LT into three groups based on sample frequency: the first three as "many", the next four as "medium", and the last three as "few". We then assessed classification performance within each group. We compared several methods under the imbalance setting of $\gamma_l = 100$ and $\gamma_u = 1$, including FixMatch/ReMixMatch, their variants augmented with SAW Lai et al. (2022) and SAW+cRT Kang et al. (2020), and our proposed FixMatch/ReMixMatch+DPC. As shown in Table 10, DPC consistently outperforms the baselines on the "few" classes, demonstrating its superiority in handling severe class imbalance.

Table 10: Fine-grained classification performance on CIFAR-10-LT ($\gamma_l = 100, \gamma_u = 1$)

| CIFAR-10-LT ($\gamma_l = 100, \gamma_u = 1$) | | | | |
|---|---|---|---|---|
| Algorithm | Overall | Many | Medium | Few |
| FixMatch | 70.2 | 96.3 | 77.7 | 34.0 |
| FixMatch+SAW | 81.2 | 95.6 | 82.9 | 64.5 |
| FixMatch+SAW+cRT | 84.6 | 87.8 | 85.5 | 80.2 |
| FixMatch+DPC | 91.1 | 95.0 | 88.6 | 93.2 |
| ReMixMatch | 65.4 | 96.6 | 70.8 | 27.0 |
| ReMixMatch+SAW | 87.0 | 96.8 | 86.4 | 78.0 |
| ReMixMatch+SAW+cRT | 88.8 | 94.5 | 87.8 | 84.4 |
| ReMixMatch+DPC | 91.0 | 95.6 | 87.9 | 90.0 |

## E   RESULTS WITH FREEMATCH AS THE BACKBONE SSL ALGORITHM

To evaluate the compatibility of DPC with recent SSL methods, we conducted experiments using FreeMatch Wang et al. (2023b) as the backbone SSL algorithm. As shown in Table 11, DPC outperformed the baselines, demonstrating its effectiveness when integrated with recent SSL frameworks.

Table 11: Comparison of bACC/GM on CIFAR-10-LT when using FreeMatch as the backbone SSL algorithm

| CIFAR-10-LT ($\gamma_l = 100$) | | |
|---|---|---|
| Algorithm | $\gamma_u = 100$ | $\gamma_u = 1$ |
| FreeMatch | 74.0/ 71.9 | 73.2/ 69.3 |
| FreeMatch+SAW+cRT | 82.8/ 82.3 | 86.4/ 86.2 |
| FreeMatch+CoSSL | 81.7/ 81.3 | 87.9/ 87.6 |
| FreeMatch+RECD | 83.8/ 83.5 | 90.8/ 90.7 |
| FreeMatch+CDMAD | 84.3/ 83.9 | 87.5/ 87.0 |
| **FreeMatch+DPC** | **84.5/ 84.2** | **91.1/ 91.0** |

## F PSEUDO-CODE

Algorithm 1 shows the pseudo-code for the training process of DPC.

---
**Algorithm 1** Pseudo-code for DPC
---
**Require:** Labeled training set $\mathcal{X}$ and unlabeled training set $\mathcal{U}$
**Ensure:** Balanced feature extractor $\xi$ and balanced classifier $\phi_{cls}$
1: Initialize $p_u^{(0)}(y) = \frac{N_k}{\sum N_k}$ for $k = 1, ..., K$
2: **while** training step $t$ **do**
3:      $\mathcal{B}_{\mathcal{X}} = \{(x_b^m, y_b^m) : b \in (1, \ldots, B_l)\} \subset \mathcal{X}$
4:      $\mathcal{B}_{\mathcal{U}} = \{(u_b^m) : b \in (1, \ldots, Bu)\} \subset \mathcal{U}$
5:      $q_{psl} = \phi_{psl}(\xi(\alpha(u_b^m))$ and $\hat{q}_{psl} = \arg\max(q_b^m)$ for $b = 1, ..., B_u$
6:      $p_u^{(t)}(y) = \lambda p_u^{(t-1)}(y) + (1 - \lambda)\frac{1}{B_u}\sum_{u_b^m \in \mathcal{B}_{\mathcal{U}}} q_{psl}$
7:      $\mathbf{W}(x_b^m) = \frac{N_K}{N_{y_b^m}}$ for $b = 1, ..., B_l$
8:      $\mathbf{W}(u_b^m) = \frac{\min p_u(y)}{p_u(y)_{\hat{q}_{psl}}}$ for $b = 1, ..., B_u$
9:      $\mathbf{M}(u_b^m) = \mathbf{I}(\max(q_{psl}) > \rho_{\hat{q}_{psl}})$ for $b = 1, ..., B_u$
10:      $L_{back}$ : Backbone loss with $(\mathcal{B}_{\mathcal{X}}, \mathcal{B}_{\mathcal{U}}, q_{psl})$
11:      $L_{psl}$ : PSL loss with $(\mathcal{B}_{\mathcal{X}}, \mathbf{W}(x_b^m))$
12:      $L_{cls}$ : CLS loss with $(\mathcal{B}_{\mathcal{X}}, \mathbf{W}(x_b^m), \mathcal{B}_{\mathcal{U}}, \mathbf{W}(u_b^m), \mathbf{M}(u_b^m), q_{psl})$
13:      $L_{DPC} = L_{back} + L_{psl} + L_{cls}$
14:      Update algorithm parameters $\theta$ using $L_{DPC}$
15: **end while**

---

## G SENSITIVITY OF HYPERPARAMETER $\eta$

We conducted additional experiments on CIFAR-10-LT by varying $\eta$, and the results, as shown below, confirm that DPC is not highly sensitive to the choice of $\eta$.

Table 12: Comparison of bACC on CIFAR-10-LT when varying $\eta$

| CIFAR-10-LT ($\gamma_l = 100$) | | |
|---|---|---|
| $\eta$ | $\gamma_u = 100$ | $\gamma_u = 1$ |
| 0.1 | 84.7 | 90.2 |
| 0.25 | 84.5 | 91.1 |
| 0.5 | 84.3 | 91.4 |
| 1.0 | 84.2 | 91.2 |

## H  T-SNE VISUALIZATIONS OF CIFAR-10-LT TEST SET FEATURES

Fig. 5 presents the T-SNE visualization on the test set features of CIFAR-10-LT under $\gamma_l = 100, \gamma_u = 1$. Fig. 5a presents the T-SNE results for FixMatch+DPC without the feature loss, while Fig. 5b presents the T-SNE results for FixMatch+DPC with the feature loss. As discussed in the main paper, the use of the feature loss mitigates the feature map bias, enabling better representations for underrepresented samples. As shown in Fig. 5b, when the feature loss is applied, each cluster becomes more compact, and the decision boundary is more clearly defined, resulting in improved performance.

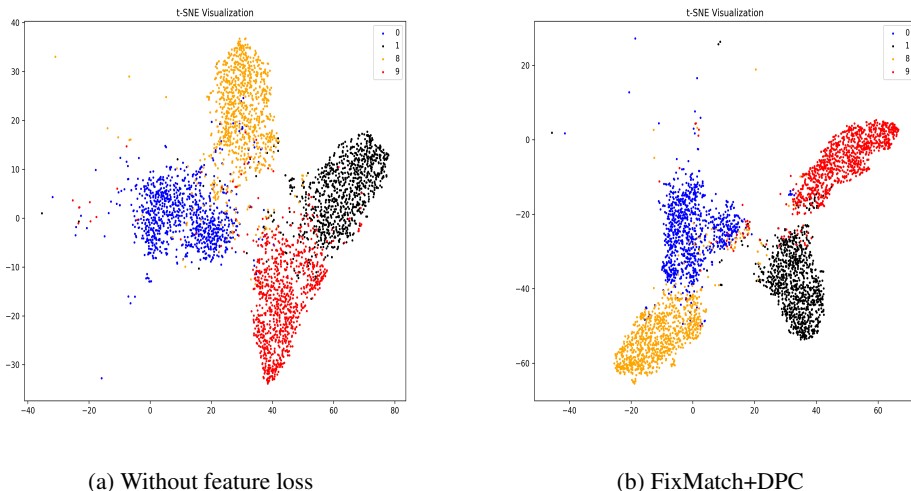

(a) Without feature loss          (b) FixMatch+DPC

Figure 5: T-SNE of the test set features of CIFAR-10-LT under $\gamma_l = 100, \gamma_u = 1$.

