# OpenReview forum: "Disentangled Pseudo-Labeling and Classification for Class-Imbalanced Semi-Supervised Learning"
_ICLR.cc/2026/Conference — Submitted to ICLR 2026_

### Official Review · Reviewer_xPhz · 2025-10-28

**Soundness:** 3
**Presentation:** 3
**Contribution:** 2
**Rating:** 6
**Confidence:** 4

**Summary:**

This paper addresses the critical challenge of class-imbalanced semi-supervised learning (CISSL).
The author(s) develop a framework that introduces an auxiliary pseudo-labeler (PSL) learned from labeled and unlabeled data to mitigate confirmation bias, together with a reweighted feature-level loss designed to alleviate representation imbalance.
Extensive experiments on small scaled long-tailed semi-supervised benchmarks demonstrate competitive or superior performance compared with existing state-of-the-art methods.

**Strengths:**

1. The paper addresses an important and well-known problem in CISSL, which is confirmation bias caused by training the classifier on both labeled and unlabeled data jointly.

2. The proposed idea is simple yet conceptually clear, and provides an intuitive way to decouple pseudo-label generation from classifier training.

**Weaknesses:**

1. The main idea of separating pseudo label generating and classifier is relatively straightforward and overlaps to some extent with prior works, such as auxiliary or debiased classifier approaches.

2. The writing style is highly uniform and repetitive,  showing a lack of natural variation and subtle reasoning. The presentation occasionally seems overly structured and mechanical, resembling auto-generated by GPT or other AI tools.

3. It remains unclear why two classifiers (PSL and CLS) are both necessary if PSL also learned from labeled data and already provides pseudo-labels. What prevents simply using PSL predictions within the backbone?
In my opinion, providing more theoretical or experimental proof will be better.
In addition, the motivation and justification for the feature-level reweighting term are not sufficiently explained.

4. Some parts of the paper, such as Eqs. (6), (8), and (9), appear overly formalized, where simple ideas are expressed through unnecessarily complex mathematical notation.

**Questions:**

1. How does DPC scale to larger and more complex datasets, such as Small-ImageNet-127 or full ImageNet-LT?

2. What is the essential benefit of separating the pseudo-labeler (PSL) and the classifier? Since both modules share the same feature extractor, how does training the PSL solely on labeled data effectively prevent confirmation bias when the shared representation is still updated using pseudo-labeled samples?

---

> ### Author Response · Authors · 2025-11-13
>
> Thank you for your thoughtful comment. We would like to respectfully clarify several points.
>
> \$W1.\$ We want to argue that our main contribution is "trainig pseudo-labeler (PSL) solely based on labeled data". Existing CISSL algorithms utilize unlabeled data in training classifiers responsible for pseudo-labeling. Then mis-predicted pseudo-labeles occur confirmation bias and further misprediction. So, we introduce PSL to avoid confirmation bias in pseudo-labeling, and the decoupling PSL from CLS (classifier for inference time) is quite natural rather than a contribution, since CLS incorporates unlabeled data to improve test performance. Our main contribution seems to be poor at glance, but this simple idea is fairly strong in that the performance of existing algorithm ABC [2] can be improved with PSL as follows.
> | Algorithm | $\gamma_{l}=\gamma_{u}=100$ | $\gamma_{l}=100, \gamma_{u}=1$ |
> |------------|:------------:|:--------------:|
> | ABC | 81.1 | 82.7 |
> | **ABC+PSL**  | **84.2** | **85.6** |
>
> \$W2.\$ We believe that the uniform tone results from our attempt to maintain a formal academic style. We will make efforts to use a more natural and varied expression in the revision.
>
> \$W3.\$ Before providing a detailed explanation, we summarize the differences between PSL and CLS in the following table.
> | Classifier | Role | Training |
> |------------|------------|--------------|
> | PSL | Generating Pseudo-label | Solely on labeled data |
> | CLS  | Inference on test samples | Both labeled and unlabeled data |
>
> Since the training set is biased towards majority classes, even if PSL provides the high-accuracy pseudo-labels, the predictions of backbone SSL algorithm also tend to be biased towards the majority class due to the inherent bias in the training set. This is the reason why we need balanced classifier CLS.
>
> If the backbone SSL loss, \$L_{back}\$, is directly reweighted by \$W\$, representation learning is disrupted and results in performance degradation as shown in [1] and [2]. To get high-quality feature and balanced classifier for inference, CLS is attatched to the representation layer of backbone SSL algorithm and it is trained by rebalanced training loss \$L_{cls}\$ based on \$W\$. PSL is trained solely on labeled data to avoid confirmation bias and predicts the class probabilities of unlabeled training samples, which are provided to \$L_{back}\$ and \$L_{cls}\$ as pseudo-labels.
>
> Furthermore, the feature map trained by backbone SSL algorithm is biased toward majority classes due to inherent bias in training set. So we mitigate feature-level imbalance using \$L_{feature}\$. Regarding the squaring of the sample weights \$W^{2}\$,  this design is intended to bias the loss toward minority classes in a manner proportional to the class imbalance in the training set. Using the weight only once would normalize all classes to the same effective weight. Squaring the weights produces a corrective loss in the opposite direction of the training set’s inherent bias, thereby alleviating feature-level imbalance.
>
> \$W4.\$ We will simplify the equation by replacing $\phi_{psl}\\left( \xi\\left( y \mid \alpha(x_b^{m}) \right) \right)$ to $f_{psl}\\left( \alpha(x_b^{m}) \right)$, and same manner for (8) and (9).
>
> \$Q1.\$ To demonstrate the effectiveness of our algorithm on a large-scale dataset, we conducted experiments on Small-ImageNet-127. Due to limited computational resources, the experiments were performed using 32×32 downsampled images. The performance results are as follows:
> | Algorithm | Averaged class recall |
> |------------|:------------:|
> | FixMatch | 29.7 |
> | FixMatch+DARP+cRT | 30.7 |
> | FixMatch+CREST++LA | 40.9 |
> | FixMatch+CoSSL | 43.7 |
> | FixMatch+RECD | 47.3 |
> | FixMatch+CDMAD | 48.4 |
> | **FixMatch+DPC** | **48.7** |
>
> \$Q2.\$ By separating the PSL and the classifier (CLS), our algorithm effectively mitigates confirmation bias at the classifier stage, allowing the CLS to be trained with more precise pseudo-labels, as shown in Table 4. The improved pseudo-label accuracy further contributes to performance enhancement. As you mentioned, the representation is updated using pseudo-labels from the PSL, then it may be influenced by the feature extractor. However, this effect is negligible compared to the direct impact of using unlabeled samples for training the classifier. Moreover, since the pseudo-label accuracy of the PSL is significantly higher than that of the backbone SSL algorithm, it actually helps alleviate feature-level imbalance and enables the generation of high-quality pseudo-labels.
>
>
> **Reference**
>
> [1] Rethinking Re-Sampling in Imbalanced Semi-Supervised Learning. In Arxiv, 2020.
>
> [2] ABC: Auxiliary Balanced Classifier for Class-Imbalanced Semi-Supervised Learning. In NeurIPS, 2021.

---

> > ### Comment · Reviewer_xPhz · 2025-11-25
> >
> > Thanks for the rebuttal. After reviewing the comments from the other reviewers and the rebuttal, I would like to maintain my initial score but lower my confidence. The additional results are based on ABC, a method proposed in 2021, which is not up to date. I will wait for further feedback from the reviewers before any additional discussion.

---

> > > ### Author Response · Authors · 2025-11-25
> > >
> > > Thank you for your response.
> > >
> > > Even if ABC is not up to date, we believe the significant performance improvement achieved when applying PSL clearly demonstrates its effectiveness. After reviewing the further feedback from the reviewers, please feel free to reach out if you have any additional questions.

---

### Official Review · Reviewer_5L6V · 2025-10-30

**Soundness:** 2
**Presentation:** 3
**Contribution:** 1
**Rating:** 2
**Confidence:** 4

**Summary:**

This paper proposes the Disentangled Pseudo-Labeling and Classification (DPC) method to address confirmation bias observed in class-imbalanced semi-supervised learning (CISSL). DPC consists of two modules: a pseudo-labeler (PSL) trained exclusively on labeled data to generate pseudo-labels, and a classifier (CLS) trained with a rebalanced loss on shared features to address feature-level imbalance. Both modules use a simple class-reweighting scheme, assigning weights inversely proportional to class frequency. Comprehensive experiments validate the superiority of the method.

**Strengths:**

+ This paper is well-motivated and easy to follow.
+ This proposed method performs well on commonly used benchmark datasets CIFAR and STL-10.

**Weaknesses:**

+ The novelty of the proposed method is limited. The main contribution of this paper lies in disentangling the pseudo-labeling and classification framework. However, the strategy of decoupling representation learning from classifier training has already been explored in recent works [1, 2], on both imbalanced supervised learning [1] and imbalanced semi-supervised learning [2]. Besides, this paper introduces an auxiliary classifier, called a pseudo-labeler, which is trained solely on labeled data to produce pseudo labels for unlabeled samples, similar to the recent method ABC [3].
+ The reweighting of sample-wise losses based on the class distribution to mitigate class imbalance is a very common practice in the CISSL setting. Similar reweighting strategies have also been employed in RECD [3] and TAF [4].
+ In addition to lacking strong technical novelty, the performance improvement appears marginal relative to the substantial computational overhead introduced by multiple forward passes. Moreover, the performance even underperforms recent methods [5, 6, 7, 8, 9].
+ Furthermore, the experimental datasets (CIFAR and STL-10) are too small to convincingly demonstrate the effectiveness of the proposed method. It would be more convincing to include evaluations on more realistic benchmarks, such as ImageNet-127 [10] or Semi-Aves [6].
+ Minor Comments:
   - In line 143, the prediction range of the model $f_{\theta}(x)$ should be $\mathbb R$, If the intention is to denote the output range of the function $\xi$, please make this distinction explicitly.
   - In Equation (6), the summation subscript M is not defined.
   - There is a typo in Line 144: "mpas" → "maps".
- - -
**Reference:**
[1] Decoupling Representation and Classifier for Long-Tailed Recognition. In ICLR, 2020.
[2] Rethinking Re-Sampling in Imbalanced Semi-Supervised Learning. In Arxiv, 2020.
[3] ABC: Auxiliary Balanced Classifier for Class-Imbalanced Semi-Supervised Learning. In NeurIPS, 2021.
[4] Rebalancing Using Estimated Class Distribution for Imbalanced Semi-Supervised Learning under Class Distribution Mismatch. In ECCV, 2024.
[5] Triplet Adaptation Framework for Robust Semi-Supervised Learning. In TPAMI, 2024.
[6] DASO: Distribution-aware semantics-oriented pseudo-label for imbalanced semi-supervised learning.
[7] Towards realistic long-tailed semi-supervised learning: Consistency is all you need. In CVPR, 2023.
[8] Twice class bias correction for imbalanced semi-supervised learning. In AAAI, 2024.
[9] SimPro A Simple Probabilistic Framework Towards Realistic Long-Tailed Semi-Supervised Learning.
[10] COSSL: Co-learning of representation and classifier for imbalanced semi-supervised learning. In CVPR, 2022.

**Questions:**

Please see the weaknesses.

---

> ### Author Response · Authors · 2025-11-13
>
> Thank you for your valuable feedback. We would like to respectfully clarify several points.
>
> \$W1.\$ We want to argue that our main contribution is "trainig pseudo-labeler (PSL) solely based on labeled data".  Existing CISSL algorithms (including [2] and [3]) utilize unlabeled data in training classifiers responsible for pseudo-labeling. Then mis-predicted pseudo-labeles occur confirmation bias and further misprediction. So, we introduce PSL to avoid confirmation bias in pseudo-labeling, and the decoupling PSL from CLS (classifier for inference time) is quite natural rather than a contribution, since CLS incorporates unlabeled data to improve test performance. Our main contribution seems to be poor at glance, but this simple idea is fairly strong in that the performance of ABC [3] can be improved with PSL as follows.
> | Algorithm | $\gamma_{l}=\gamma_{u}=100$ | $\gamma_{l}=100, \gamma_{u}=1$ |
> |------------|:------------:|:--------------:|
> | ABC | 81.1 | 82.7 |
> | **ABC+PSL**  | **84.2** | **85.6** |
>
> \$W2.\$ To specify our main idea PSL, we constructed simple rebalancing framework. Many of recent CISSL algorithms are based on Logit-Adjustment (LA) due to its strong performance. However, we want to show the effectiveness of PSL by combining simple rebalancing strategy, reweighting based on class distribution.  Furthermore, to address feature-level bias, we introduce the feature loss \$L_{feature}\$, so we utilized the class weight \$W\$ in more various way compared to RECD[3] and TAF[4].
>
> \$W3.\$ Firstly, our multiple heads rarely adds computational cost as follows. We measured the FLoating point OPerations per Second (FLOPS) for FixMatch, FixMatch+DPC, ReMixMatch, and ReMixMatch+DPC using an NVIDIA RTX 3090 GPU. The proposed additional module and feature loss resulted in a modest decrease in training speed (approximately 7–10%), yet this overhead is negligible in our opinion.
> | Algorithm | iteration/sec |
> |------------|:------------:|
> | FixMatch | 17.75 |
> | FixMatch+DPC  | 16.02 |
> | ReMixMatch | 6.17 |
> | ReMixMatch+DPC  | 5.73 |
>
> Furthermore, we compared our method with DASO [6] and ACR [7] in the Appendix (after the references in the paper), and our method outperforms them across all settings. We excluded TAF [5] and TCBC [8] because their public source code is not available, making direct comparison difficult since the performance of many baselines differs. Additionally, we excluded SimPro [9] as the publicly available code does not seem to reproduce the reported performance. Overall, our algorithm outperforms the existing CISSL methods for which a fair comparison is possible.
>
> \$W4.\$ To demonstrate the effectiveness of our algorithm on a large-scale dataset, we conducted experiments on Small-ImageNet-127. Due to limited computational resources, the experiments were performed using 32×32 downsampled images. The performance results are as follows:
> | Algorithm | Averaged class recall |
> |------------|:------------:|
> | FixMatch | 29.7 |
> | FixMatch+DARP+cRT | 30.7 |
> | FixMatch+CREST++LA | 40.9 |
> | FixMatch+CoSSL | 43.7 |
> | FixMatch+RECD | 47.3 |
> | FixMatch+CDMAD | 48.4 |
> | **FixMatch+DPC** | **48.7** |
>
> \$MC1.\$ Thank you for your correction. We will fix it by "that accurately predicts the class labeles of test samples" --> "that accurately predicts the class probabilities of test samples" in line 143.
>
> \$MC2, MC3. \$ Thank you for pointing them out. In Section 3.1, we denoted the minibatch as \$B_{x}\$ so \$MB_{x}\$ is indeed a typo. We will correct them in the final version.

---

### Official Review · Reviewer_E2LY · 2025-10-30

**Soundness:** 3
**Presentation:** 3
**Contribution:** 3
**Rating:** 6
**Confidence:** 3

**Summary:**

This paper introduces Disentangled Pseudo-Labeling and Classification (DPC) for class-imbalanced semi-supervised learning. DPC separates pseudo-label generation from classifier training to reduce confirmation bias and employs reweighted and feature-level losses to handle imbalance.

**Strengths:**

**Originality**: The paper presents a clear and original idea by explicitly disentangling pseudo-label generation from classifier training, effectively addressing the confirmation bias issue in class-imbalanced semi-supervised learning.

**quality**: The experimental quality is strong, with comprehensive comparisons, ablation studies, and sensitivity analyses on multiple benchmarks.

**Clarity**: The structure of this paper is clear, and the language is well-articulated.

**Significance**: The work provides a significant and practical contribution to improving robustness and fairness in semi-supervised learning under class imbalance.

**Weaknesses:**

1.The robustness of the pseudo-labeling module (PSL) under extreme imbalance or very few labeled samples is not fully analyzed—PSL may overfit when labeled data are highly limited.

2.The EMA-based estimation of unlabeled data distribution could be unstable when the unlabeled set is highly biased, yet the paper lacks a sensitivity or robustness analysis for such cases.

3.The masking rule (“top 50% most frequent classes” and class-specific threshold ρₖ) appears empirical. The authors should clarify the rationale for these design choices and provide sensitivity studies

**Questions:**

1.Both the masking rule and the class reweighting scheme are crucial to DPC’s success, yet their design choices are largely empirical and lack sufficient analysis.

2.Since the initial unlabeled distribution $p_u^{0}$​ is initialized from the labeled set, how would DPC behave if the labeled data were biased?

3.The notation of the subscript $x_b^{m} \in MB_{x}$​ in Equation (6) is ambiguous and requires clarification.

4.The expression for $\gamma_{u}$ on line 139 is not entirely correct and could easily confuse readers. $M_k$ is the number of unlabeled samples of class k. However, $\gamma_{u}$ should be calculated based on the $M_k$ sequence.

---

> ### Author Response · Authors · 2025-11-13
>
> Thank you for your valuable feedback. We would like to respectfully clarify several points.
>
> \$W1.\$ We understand the reviewer’s concern that relying solely on labeled data for pseudo-labeling may raise questions about robustness under extreme imbalance or when only a few labeled samples are available. However, in such scenarios, assigning reliable pseudo-labels to unlabeled samples from minority classes becomes increasingly difficult. The pseudo-labels tend to be biased toward majority classes, which accelerates confirmation bias without effectively addressing overfitting.
>
> To further support this point with experimental results, please refer to Table 2 and Table 4 in the paper. We conducted experiment on CIFAR-100-LT under the setting \$\gamma=100\$, where some classes hae only one labeled sample. Our model produces much more precise pseudo-labels with PSL, resulting in improved test performance.
>
> Furthermore, we conducted additional experiments under more extreme scarcity conditions. Specifically, we modified the CIFAR-10-LT setting by reducing the number of labeled samples for the most frequent class \$(N_{1})\$ from \$1500\$ to \$500\$ and \$100\$. In the most extreme case \$(N_{1}=100)\$, the labeled sample counts per class are \$[100,59,35,21,12,7,4,2,1,1]\$. We compared the performance of DPC with recent state-of-the-art CISSL algorithms, CDMAD and RECD. The results show that DPC consistently outperforms these methods, especially in the presence of minority classes with very few labeled samples but many unlabeled ones. These findings suggest that DPC is more robust to confirmation bias and remains effective even with severely limited labeled data.
> | Algorithm | $\gamma_{l}=\gamma_{u}=100$ | $\gamma_{l}=100, \gamma_{u}=1$ |
> |------------|:------------:|:--------------:|
> | CDMAD | 78.9 | 80.2 |
> | RECD  | 78.0 | 85.7 |
> | **DPC** | **79.5** | **89.3** |
>
> | Algorithm |$\gamma_{l}=\gamma_{u}=100$ | $\gamma_{l}=100, \gamma_{u}=1$|
> |------------|:------------:|:--------------:|
> | CDMAD | 63.2 | 70.7 |
> | RECD  | 59.1 | 70.9 |
> | **DPC** | **63.9** | **80.2** |
>
> \$W2.\$ To assess the stability of the EMA-based estimation under highly biased unlabeled sets, we measured the KL divergence between the estimated unlabeled class distribution and the true class distribution for each of the last 100 epochs. We experimented on CIFAR-10-LT under the setting \$\gamma_{l}=\gamma_{u}=150\$. The minimum, maximum, mean, and standard deviation of the KL divergence were 0.00021, 0.00049, 0.00033, and 0.00007, respectively. These results indicate that the estimation remains highly stable across all epochs, even when the unlabeled data are strongly biased. We attribute this stability to PSL, which consistently provides relatively accurate pseudo-labels throughout training.
>
> \$W3, Q1.\$ Firstly, we would like to clarify that the reweighting scheme is based on the class distribution of the training set. While this represents a simple and fundamental strategy for addressing class imbalance, our experimental results demonstrate that it can become particularly effective when combined with the PSL module, as implemented in DPC.
>
> Secondly, we observed that mis-predicted pseudo-labels often exhibit lower confidence compared to correctly predicted pseudo-labels, and the average confidence tends to vary across different classes. To account for this, we set the threshold \$\rho_{k}\$ to the mean confidence for each class \$k\$.
>
> Lastly, we acknowledge that setting "top \$K\$% most frequent classes" is somewhat empirical. However, in practice, the choice of $K$ can vary over a wide range, and the model's performance is not highly sensitive to this value as follows.
> | \$K\$ | $\gamma_{l}=\gamma_{u}=100$ | $\gamma_{l}=100, \gamma_{u}=1$ |
> |------------|:------------:|:--------------:|
> | 20 | 84.7 | 90.8 |
> | 30  | 84.6 | 90.6 |
> | 40  | 84.3 | 91.0 |
> | 50 | 84.5 | 91.1 |
>
> \$Q2.\$ Since one of the goals of the algorithm is to mitigate bias and recover the true distribution, the impact of the initialization on the final outcome is minimal. In fact, all of our experiments were conducted under settings where the labeled data were biased. As shown in Figure 4, DPC accurately estimated the class distribution of the unlabeled data regardless of whether the labeled and unlabeled class distributions were matched or mismatched.
>
> \$Q3.\$ Thank you for pointing this out. In Section 3.1, we denoted the minibatch as \$B_{x}\$ so \$MB_{x}\$ is indeed a typo. We will correct it in the final version.
>
> \$Q4.\$ As mentioned in line 137, we did not assume any ordering among \$M_{k}\$. Therefore, \$\gamma_{u}\$ was set as the ratio between the largest and smallest values of \$M_{k}\$. Note that this value was defined solely to illustrate our experimental setting and was not used as a prior during the actual experiments.

---

### Official Review · Reviewer_ft2t · 2025-10-30

**Soundness:** 2
**Presentation:** 2
**Contribution:** 2
**Rating:** 4
**Confidence:** 5

**Summary:**

This paper aims to address confirmation bias in class-imbalanced semi-supervised learning (CISSL). The authors argue that confirmation bias arises because the classifier that generates pseudo-labels is simultaneously trained on the unlabeled data it labels, forming a self-reinforcing loop. To break this loop, the paper proposes the DPC framework, whose core idea is to disentangle pseudo-label generation from the final classification task. DPC introduces a “pseudo-labeler” (PSL) trained only on labeled data to produce higher-quality pseudo-labels, and a “classifier” (CLS) that is trained using these pseudo-labels. In addition, DPC designs a feature loss to mitigate imbalance at the representation level. The authors conduct extensive experiments on multiple CISSL benchmarks.

**Strengths:**

- Adding two heads (PSL/CLS) and loss terms on top of an existing SSL backbone entails minimal engineering changes and offers good reusability.
- The authors perform comprehensive experiments on standard datasets (CIFAR-10-LT, CIFAR-100-LT, STL-10-LT) and across both distribution-matched and mismatched imbalance settings, comparing against many recent baselines. The experimental coverage is broad and the comparisons are thorough.
- CISSL is a real and highly challenging core problem, making the topic timely and important.

**Weaknesses:**

- Decoupling or adding auxiliary modules in class-imbalanced learning is a common strategy; methodologically, the innovative contribution of this paper is limited.

- The DPC framework integrates a backbone SSL algorithm, PSL, and CLS—three classification modules—plus multiple losses (L_back, L_psl, L_cls, L_feature). The system is overly complex. The intricate interactions among modules (e.g., both L_back and L_feature update the shared feature extractor) are not clearly explained or analyzed, giving the impression of a carefully tuned ensemble rather than a simple, elegant solution. The authors should further clarify the role of each module and how the losses are balanced. There are additional ambiguities that need explanation, such as the difference between L_back and L_psl, their respective purposes, why both are necessary, and which classifier is used at inference time.

- The loss functions rely on sample-wise reweighting based on W. By contrast, many SOTA methods use Logit Adjustment (LA)-based losses, which have stronger theoretical foundations and, in my experience, outperform sample-wise weighting. This has also been discussed in many supervised learning papers. Please justify why sample-wise weighting is preferable in the semi-supervised setting. Moreover, the paper does not provide code; please release reproducible open-source code during rebuttal.

- The paper claims "DPC also addresses feature representation imbalance," yet it is widely accepted in the long-tailed literature (e.g., Decoupling representation and classifier for long-tailed recognition) that reweighting harms representation learning. The authors should explain why reweighting here improves representation learning. In addition, the motivation for squaring the sample weights (W^2) is vague and ad hoc. Why square rather than use, for example, 1.5-power or 3-power? This key design lacks theoretical or strong empirical support.

- In Table 5, removing the core PSL module (“Without PSL”) on STL-10-LT (γl=10) only drops performance from 81.6 to 80.9. Such a small decrease severely weakens the central claim that “PSL is key to solving confirmation bias.” If a complex PSL module yields only a negligible gain, its necessity is questionable. Similarly, removing the feature loss (“Without feature loss”) or masking (“Without masking”) results in very small drops, suggesting that the practical contributions of these components may be limited.

- Although the method is claimed to be SOTA, some presented results do not surpass existing SOTA approaches; under certain settings, performance is worse than ACR (published two years ago).

**Questions:**

Refer to weaknesses.

---

> ### Author Response · Authors · 2025-11-12
>
> Thank you for your thoughtful comment. We would like to respectfully clarify several points.
>
> \$W1.\$ We agree that decoupling or adding auxiliary modules is a common strategy in class-imbalanced learning. However, our main contribution is not the decoupling. Our core idea is to train the pseudo‑labeling classifier (PSL) only on labeled data, which mitigates confirmation bias at the classifier stage caused by mis‑predicted unlabeled samples. The decoupling of PSL from the CLS is simply a design choice to enable this objective, rather than the main novelty of the method. We appreciate the opportunity to clarify this point and will revise the manuscript to make our contribution statement explicit.
>
> \$W2.\$ We acknowledge that our model may appear complex. Conceptually, the structure is relatively straightforward: a backbone SSL algorithm provides the representation layer, to which two classifiers are attached—PSL, responsible for pseudo-labeling, and CLS, used for inference.
>
> The losses of PSL and CLS are reweighted based on class-specific weights \$W\$ derived from the class distribution. We do not directly reweight the backbone SSL loss (\$L_{back}\$) because doing so can destabilize representation learning. Instead, to mitigate feature-level imbalance, we derive a feature loss \$L_{feature}\$ from \$L_{cls}\$ and backpropagate it to the representation layer, thereby enhancing the quality of learned features.
>
> PSL and CLS are kept separate to handle unlabeled samples effectively. Using unlabeled data for classifier training can improve inference performance; however, directly training on pseudo-labeled samples risks confirmation bias from mispredictions. Therefore, PSL is trained exclusively on labeled data to generate reliable pseudo-labels, while CLS leverages both labeled and pseudo-labeled data to optimize performance at inference time.
>
> In this way, each module has a clear and distinct role: PSL ensures reliable pseudo-labeling, CLS improves inference accuracy by utilizing both labeled and unlabeled data, and \$L_{feature}\$ guides balanced representation learning of the backbone.
>
> \$W3.\$ We acknowledge that LA-based losses are theoretically stronger than sample-wise reweighting and that many recent SOTA methods adopt LA-based approaches. However, this actually highlights the effectiveness of PSL in our framework. Despite using the simple technique of sample-wise reweighting, the combination with PSL allows our model to achieve competitive or even superior performance compared to recent LA-based SOTA methods such as CDMAD and ACR.
>
> Moreover, reweighting is necessary in our framework because \$L_{feature}\$ is derived based on class distribution, and combining it with sample-wise reweighting effectively mitigates feature-level imbalance without disrupting representation learning.
>
> Finally, we have included the appendix and source code in the supplementary material for reproducibility, and we encourage reviewers to refer to them.
>
> \$W4.\$ We agree that direct reweighting of the backbone SSL algorithm can harm representation learning, as widely observed in the long-tailed literature. Therefore, we do not reweight the backbone loss; the backbone is trained without rebalancing. Instead, to mitigate feature-level bias, we introduce \$L_{feature}\$, derived from \$L_{cls}\$.
>
> Regarding the squaring of the sample weights \$W^{2}\$, this design is intended to bias the loss toward minority classes in a manner proportional to the class imbalance in the training set. Using the weight only once would normalize all classes to the same effective weight. Squaring the weights produces a corrective loss in the opposite direction of the training set’s inherent bias, thereby alleviating feature-level imbalance.
>
> \$W5.\$ While it may appear negligible, we would like to provide additional context. As shown in Table 3, the STL-10-LT performance of CDMAD (CVPR 2024) and DARP+cRT (NeurIPS 2020) differs by only 0.6~1.1 despite a span of several years of algorithmic development. In this context, the 0.7 drop observed without PSL is meaningful and non-trivial. A similar interpretation applies to the feature loss and masking components.
>
> Moreover, Table 4 demonstrates that, under certain settings, removing PSL can lead to a performance drop of up to 1.8, further confirming that PSL is an effective and important module in mitigating confirmation bias. Therefore, we believe that the contributions of PSL, feature loss, and masking, while individually modest, collectively play a significant role in achieving strong performance.
>
> \$W6.\$ Since the experimental settings used to evaluate DASO and ACR differ slightly from ours, we refrained from directly comparing their reported performance with our results. To ensure a fair comparison, we evaluated DASO and ACR under the same experimental configurations as DPC. As shown in Tables 6,7,8,9 in Appendix, DPC consistently outperformed both DASO and ACR.

---

### Meta-Review · Area_Chair_qzYo · 2025-12-12

**Summary:**

Reviewers raised a range of concerns about the paper’s novelty, clarity, complexity, and empirical validation. Across reviews, a core concern is limited methodological novelty: disentangling pseudo-labeling from classifier training and using auxiliary classifiers or reweighting strategies have been explored in prior CISSL and long-tailed learning work, and several reviewers felt the contribution is incremental relative to existing methods. Reviewers also questioned the necessity and complexity of the framework, noting that DPC combines multiple heads and losses with interactions that are not always clearly motivated or analyzed, giving the impression of a carefully tuned system rather than a simple, principled solution.

Several concerns focused on design choices and theoretical grounding. The reliance on sample-wise reweighting instead of more established logit-adjustment losses was questioned, as was the motivation for specific heuristics such as squaring class weights, EMA-based estimation of unlabeled distributions, and masking rules; these choices were seen as empirical and insufficiently justified. Reviewers further expressed skepticism about whether components like the pseudo-labeler, feature loss, and masking are truly beneficial, since ablation results often show only modest performance drops when they are removed.

On the empirical side, reviewers noted that performance gains over strong baselines are sometimes small or inconsistent, with some settings underperforming prior methods. The evaluation was also criticized for relying mainly on small-scale benchmarks (CIFAR and STL-10), raising doubts about scalability to larger or more realistic datasets.

**Reviewer Concerns:**

Some reviewer concerns were at least partially addressed in the rebuttal. The authors clarified the intended novelty and positioning of DPC relative to prior CISSL methods, emphasizing the strict separation between pseudo-label generation and classifier training and arguing that this separation directly targets confirmation bias. They also provided additional justification for several design choices—such as training the pseudo-labeler only on labeled data, propagating a reweighted feature loss to the backbone, and using EMA to estimate unlabeled class distributions—and pointed to ablation studies as evidence that each component contributes to performance.

However, several concerns remain outstanding even after the rebuttal. Questions about methodological novelty are only partially resolved: while the rebuttal clarifies how DPC differs from prior auxiliary-classifier approaches, some reviewers may still view the contribution as incremental, as it relies on familiar ingredients (reweighting, auxiliary heads, masking) combined in a new way rather than introducing a fundamentally new principle. Concerns about complexity and over-engineering also persist, since the rebuttal largely defends the necessity of each component without offering a simpler variant or stronger theoretical justification for why these particular heuristics are optimal.

On the empirical side, key limitations remain. The rebuttal does not substantially address concerns about evaluation scope, as experiments remain limited to CIFAR and STL benchmarks, leaving open questions about scalability to larger datasets or more realistic settings. Similarly, while ablations are cited, the fact that performance drops are often modest leaves it unclear whether all components are essential.

**Reviewer Scores:**

Reviewer ft2t (score 4) raised concerns about limited novelty, over-complexity, unclear roles of multiple losses and classifiers, weak ablation evidence for key components, and questionable design choices (e.g., sample-wise reweighting, squared weights). The rebuttal directly addressed many of these points by clarifying the intended contribution (PSL being trained only on labeled data), explaining the role of each module, adding sensitivity analyses, and defending the significance of modest gains in the CISSL context. However, the rebuttal does not fundamentally resolve the concern about novelty or the necessity of multiple components, nor does it resolve concerns raised by ablations showing small changes. Therefore, it is plausible that the reviewer would maintain the initial score.

Reviewer E2LY (score 6) was generally positive, with concerns focused on robustness under extreme imbalance, EMA stability, empirical heuristics (masking and reweighting), and minor technical clarity issues. The rebuttal addressed these points, providing additional experiments under extreme label scarcity, stability analyses for EMA estimation, and sensitivity studies for masking thresholds, as well as clarifications of notation errors. Given that this reviewer was already inclined toward acceptance and expressed some uncertainty, the score would likely remain positive.

Reviewer 5L6V (score 2) was concerned about a lack of novelty, marginal gains relative to computational overhead, underperformance compared to recent methods, and insufficient evaluation on large-scale datasets. While the rebuttal attempted to address these points by reframing the contribution, reporting FLOPs and training speed, adding Small-ImageNet-127 results, and arguing about fairness of comparisons, the reviewer’s critique is largely conceptual rather than based on missing clarification. Given the low initial score and strong stance, it is very unlikely that the score would change.

Reviewer xPhz (score 6) criticized the writing style and over-formalized presentation, and questioned the necessity of having both PSL and CLS. The rebuttal addressed these points directly by clarifying the distinct roles of PSL and CLS, explaining why PSL predictions alone are insufficient, simplifying equations, adding large-scale experiments, and acknowledging stylistic issues. However, the reviewer explicitly followed up after the rebuttal to state that considering the comments of the other reviewers they would maintain their initial score but lowering their confidence. Moreover, the reviewer noted that additional results relied on an older baseline (ABC) and that novelty concerns remained.

Overall, no reviewer appears likely to have changed their score as a result of the rebuttal, with one positive reviewer explicitly indicating reduced confidence after discussion. The Area Chair acknowledges the rebuttal and the additional clarifications and experiments, but ultimately agrees with the reviewers’ concerns regarding incremental novelty, limited evaluation scope, and the lack of consistently strong improvements over prior work.

---

### Decision · Program_Chairs · 2026-01-26

Reject